# Lineage-tracing and translatomic analysis of damage-inducible mitotic cochlear progenitors identifies candidate genes regulating regeneration

Tomokatsu Udagawa[1,2°], Patrick J. Atkinson[1°], Beatrice Milon[3], Julia M. Abitbol[1], Yang Song[4], Michal Sperber[5], Elvis Huarcaya Najarro[1], Mirko Scheibinger[1], Ran Elkon[5], Ronna Hertzano[3,4,6], Alan G. Cheng[1]*

**1** Department of Otolaryngology-Head and Neck Surgery, Stanford University School of Medicine, Stanford, California, United States of America, **2** Department of Otorhinolaryngology, The Jikei University School of Medicine, Tokyo, Japan, **3** Department of Otorhinolaryngology-Head and Neck Surgery, University of Maryland School of Medicine, Baltimore, Maryland, United States of America, **4** Institute for Genome Sciences, University of Maryland School of Medicine, Baltimore, Maryland, United States of America, **5** Department of Human Molecular Genetics and Biochemistry, Sackler School of Medicine, Tel Aviv University, Tel Aviv, Israel, **6** Department of Anatomy and Neurobiology, University of Maryland School of Medicine, Baltimore, Maryland, United States of America

° These authors contributed equally to this work.
* aglcheng@stanford.edu

**Data Availability Statement:** All relevant data are within the paper and its Supporting Information files.

## Abstract

Cochlear supporting cells (SCs) are glia-like cells critical for hearing function. In the neonatal cochlea, the greater epithelial ridge (GER) is a mitotically quiescent and transient organ, which has been shown to nonmitotically regenerate SCs. Here, we ablated Lgr5+ SCs using *Lgr5-DTR* mice and found mitotic regeneration of SCs by GER cells in vivo. With lineage tracing, we show that the GER houses progenitor cells that robustly divide and migrate into the organ of Corti to replenish ablated SCs. Regenerated SCs display coordinated calcium transients, markers of the SC subtype inner phalangeal cells, and survive in the mature cochlea. Via RiboTag, RNA-sequencing, and gene clustering algorithms, we reveal 11 distinct gene clusters comprising markers of the quiescent and damaged GER, and damage-responsive genes driving cell migration and mitotic regeneration. Together, our study characterizes GER cells as mitotic progenitors with regenerative potential and unveils their quiescent and damaged translatomes.

## Introduction

The cochlea requires both sensory hair cells (HCs) and nonsensory supporting cells (SCs) for sound reception. HCs are mechanoreceptors that convert sound into neural impulses [1,2], and SCs mediate spontaneous calcium transients, regulate glutamate uptake, govern HC innervation, provide trophic support for spiral ganglia neurons, and participate in intercellular metabolic coupling critical for organ maturation and maintenance [3–8].

**Funding:** This work was supported by the Lucile Packard Foundation for Children's Health, Stanford NIH-NCATS-CTSA UL1 TR001085, Garnett Passe and Rodney Williams Memorial Foundation Research Training Grant (P.J.A.), Stanford School of Medicine Dean's postdoctoral fellowship (J.M. A.), Child Health Research Institute of Stanford University (E.H.N.), NIDCD/NIH R01DC013817, Department of Defense MR130240 (R.H.), NIH/ NIDCD K08DC011043, RO1DC01910, RO1DC13910, Department of Defense MR130316, Akiko Yamazaki and Jerry Yang Faculty Scholar Fund, California Initiative in Regenerative Medicine RN3-06529, and Yu and Oberndorf families (A.G. C.). The funders had no role in study design, data collection and analysis, decision to publish, or preparation of the manuscript.

**Competing interests:** The authors have declared that no competing interests exist.

**Abbreviations:** ABR, auditory brainstem response; CPM, count per kilobase million; CQN, conditional quantile normalization; DC, Deiters' cell; DEG, differentially expressed gene; DPOAE, distortion product otoacoustic emission; DTR, diphtheria toxin receptor; EF, enrichment factor; GER, greater epithelial ridge; HBSS, Hanks' Balanced Salt Solution; HC, hair cell; IHC, inner hair cell; IP, immunoprecipitation; IPhC, inner phalangeal cell; LER, lesser epithelial ridge; OCT, optimal cutting temperature; OHC, outer hair cell; PBS, phosphate buffered solution; PCA, principal component analysis; PFA, paraformaldehyde; RIN, RNA integrity number; SC, supporting cell; SPL, sound pressure level.

In nonmammalian vertebrates, SCs also act as transit amplifying cells and HC precursors by proliferating and regenerating lost HCs [9–11]. By contrast, the mature mammalian cochlea neither regenerate nor proliferate, thus both HC and SC degenerations lead to permanent hearing loss. While the neonatal cochlea harbors SCs, particularly those expressing the Wnt target gene *Lgr5*, which are capable of proliferating and regenerating lost HCs, they are limited in numbers and spatially restricted, with regenerated cells being short-lived [12–14]. One prior study showed nonmitotic regeneration of the SC subtype inner phalangeal cells (IPhCs) following selective ablation, and that broadening the extent of ablation to include the greater epithelial ridge (GER) reduced regeneration, suggesting the presence of SC precursors therein [15].

Here, we selectively ablated Lgr5$^+$ SCs using the *Lgr5$^{DTR/+}$* mouse line [16] and found robust proliferation in the GER in the neonatal cochlea. As a transient structure during neonatal stages of cochlear development [17], the GER's role in development is incompletely understood [18]. We fate-mapped GLAST-Cre$^+$ GER cells in vivo and observed that they proliferated and migrated to replenish IPhCs. Regenerated cells expressed markers of nascent and mature IPhCs, displayed spontaneous calcium activity, and remained present in the mature cochlea. Using GLAST-RiboTag mice, a method to enrich transcripts in the GER region, we identified differentially expressed genes (DEGs) upon depletion of Lgr5$^+$ cells. Together, our results show that severe cochlear SC loss stimulates mitotic regeneration by GER cells, thereby providing a framework that may guide regeneration of the mammalian cochlea.

## Results

### Supporting cell regeneration after ablation of Lgr5$^+$ cells

Sensory HCs and SCs in the cochlea are marked by expression of Myosin7a and Sox2, respectively [19,20]. SC subtypes can be grouped as those residing in the medial (GER and IPhCs) and lateral compartments (pillar and Deiters' cells (DCs)) (**S1A Fig**) [21]. To establish a model of SC ablation, we first examined cochleae from untreated *Lgr5$^{DTR-EGFP/+}$* (*Lgr5-DTR*) mice [16]. This is a knock-in model in which human diphtheria toxin receptor (DTR) and EGFP are driven by endogenous *Lgr5* expression, and DT administration induces rapid degeneration of Lgr5$^+$ cells [16,22]. In the *Lgr5-DTR* cochlea, we found organized rows of Myosin7a$^+$ outer and inner hair cells (OHCs and IHCs) intercalated by Sox2$^+$ SCs (**Fig 1A**). At postnatal day (P) 1, Lgr5-EGFP expression is restricted to specific SC subtypes (the third row of DCs, inner pillar cells, IPhCs, and lateral GER) (**Fig 1A-A' , S1A Fig**). These data corroborated previous *in situ* hybridization in wild-type cochlea and GFP expression in *Lgr5$^{EGFP-CreERT2/+}$* knock-in cochleae [23].

To selectively ablate Lgr5$^+$ SCs in the neonatal cochlea, we treated P1 *Lgr5-DTR* mice with DT (4 ng/g, IM or IP) and harvested cochleae 1 to 20 days later (**Fig 1B**). In control cochlea (saline-treated *Lgr5$^{DTR/+}$* or DT-treated wild-type mice), we did not detect any cell loss and rarely observed pyknotic nuclei among DCs, pillar cells, or IPhCs from P4 to P21 (**Fig 1C, 1G and 1I, S2A, S2C, S2E, S2G, S2I and S2K Fig, S1 and S2 Tables**). In the P4 DT-treated *Lgr5$^{DTR/+}$* cochlea, we found loss of Sox2$^+$ SCs and EGFP expression in both the lateral and medial compartments in all 3 turns, with degeneration most severe among DCs, pillar cells (both >60%), and IPhCs (>50%) (**Fig 1D and 1L, S2B and S2D Fig, S1 Table**). SC loss in the lateral compartment was preceded by ectopic EGFP expression in the first and second row of DCs and outer pillar cells (**S1B–S1I Fig**). The IPhCs are defined as 2 rows of Sparcl1$^+$ SCs medial and subjacent to the inner HCs (**Fig 1C and 1E**) [15,24]. After DT-induced damage,

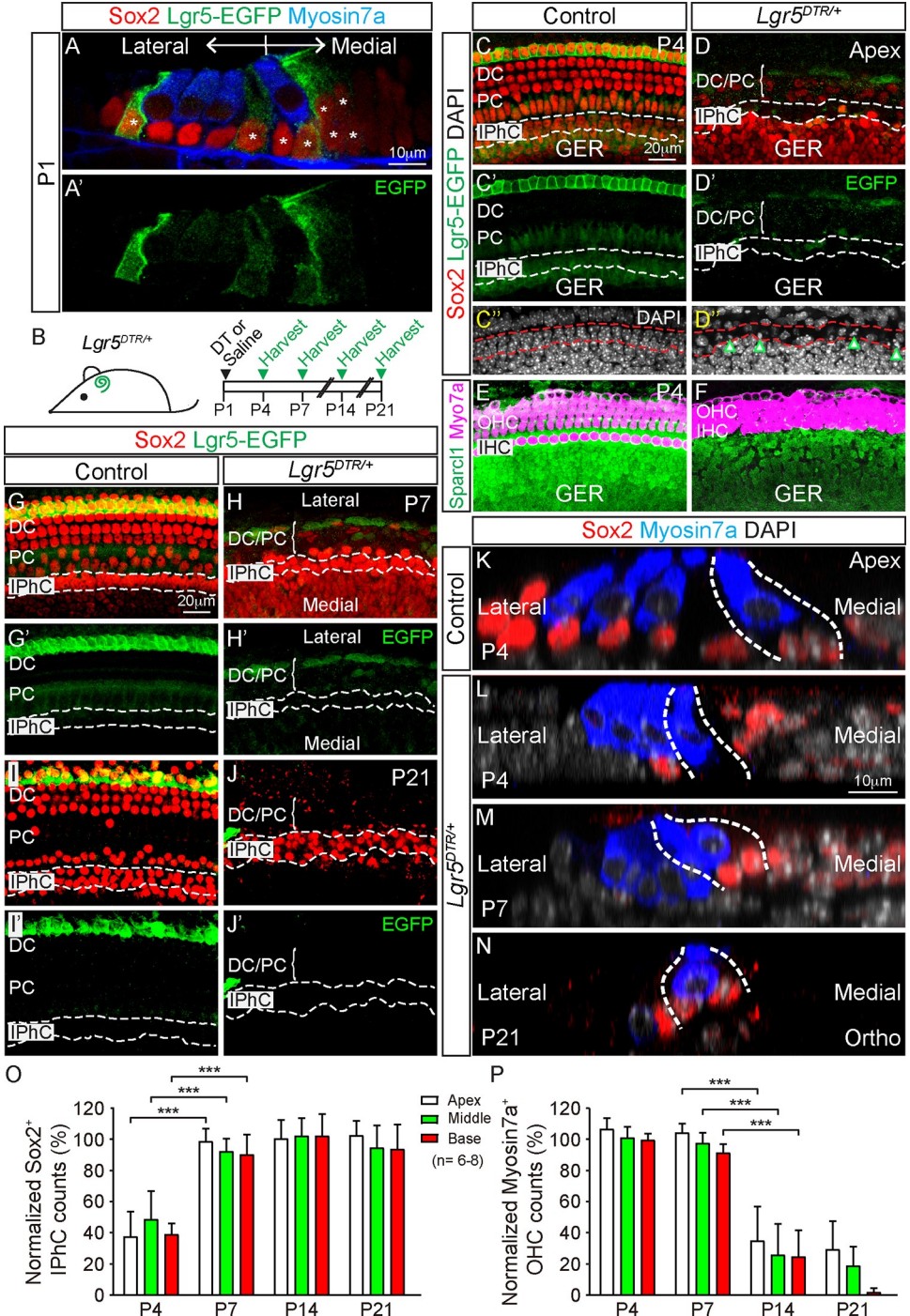

**Fig 1. IPhCs regenerate after ablation of Lgr5⁺ cells.** (A) Lgr5-EGFP expression in the lateral GER, IPhCs, inner PCs, and the third row of DCs in the P1 *Lgr5^{DTR-EGFP/+}* cochlea. (B) Schematic of the experimental paradigm: DT or saline was given to P1 *Lgr5^{DTR/+}*, and cochleae were examined at designated ages. (C) Saline-treated *Lgr5^{DTR/+}* (control) cochleae showing Lgr5-EGFP expression in a subset of Sox2⁺ SCs at P4. No pyknotic nuclei were detected in the IPhC region, which is outlined by dashed lines. For cochleae aged P4 to P7, IPhCs were identified as 2 rows of Sox2⁺ cells with nuclei immediately medial to and below IHCs. Representative images of the apical turn are shown. (D) In the P4 DT-treated *Lgr5^{DTR/+}* cochlea, there was a loss of Lgr5-EGFP⁺ cells and Sox2⁺ SCs in both the PC/DC and IPhC regions. Many pyknotic nuclei (arrowheads) were found in the IPhC region at P4. (E, F) P4 control cochlea with Sparcl1 expression in IPhCs and GER cells. Loss of Sparc1 expression in the IPhC region in the DT-treated *Lgr5^{DTR/+}* cochlea. (G, I) Saline-treated *Lgr5^{DTR/+}* (control, P7) cochleae showed Lgr5-EGFP expression in IPhCs and the third row of DCs. Lgr5-EGFP expression is limited to the third row of DCs at P21. (H, J) At P7, Sox2⁺ IPhCs were

replenished, but only few Sox2$^+$ SCs in the PC/DC region remained in the DT-treated $Lgr5^{DTR/+}$ cochlea. Similarly, many Sox2$^+$ IPhCs remained in the P21 DT-treated $Lgr5^{DTR/+}$ cochlea, while few Sox2$^+$ SCs were found in PC/DC region. (K-N) Orthogonal views of control cochlea with Sox2$^+$ SCs and Myosin7a$^+$ HCs at P4 cochlea (K). In the P4 DT-treated $Lgr5^{DTR/+}$ cochlea, there was loss of SCs including IPhCs (L). Progressive loss of Myosin7a$^+$ OHCs in the lateral compartment and regeneration of IPhCs and survival of IHCs in the medial compartment (M, N). (O) Normalized Sox2$^+$ IPhC counts (per 160 μm) of DT-treated $Lgr5^{DTR/+}$ cochleae showing a significant loss at P4 followed by regeneration by P7 in each turn ($n = 6$ at P4, $n = 8$ at P7, $n = 6$ at P14, and $n = 8$ at P21). (P) There were significant losses of OHCs in all 3 turns of the DT-treated $Lgr5^{DTR/+}$ cochleae between P7 and P14. DT-treated wild-type and saline-treated $Lgr5^{DTR/+}$ mice served as undamaged controls. Data represent mean ± SD. $^{**}p < 0.01$, $^{***}p < 0.001$. Two-way ANOVA with Tukey's multiple comparisons test. $n = 4$–8. See S1 Data for O and P. DC, Deiters' cell; DT, diphtheria toxin; GER, greater epithelial ridge; HC, hair cell; IHC, inner hair cell; OHC, outer hair cell; IPhC, inner phalangeal cell; PC, pillar cell; SC, supporting cell.

we detected many pyknotic nuclei and loss of Sparcl1 expression in the IPhC region, indicating cell death in all 3 cochlear turns at P4 (**Fig 1D'' and 1F**, **S2B'' and S2D'' Fig**, **S2 Table**).

Three days later (P7), Sox2$^+$ IPhCs along the whole length of DT-treated $Lgr5^{DTR/+}$ cochleae significantly increased and were replenished to control levels (**Fig 1H, 1M and 1O**, **S2F and S2H Fig**, **S1 Table**). On the other hand, Sox2$^+$ SCs in the lateral compartment (DCs and pillar cells) modestly increased only in the apical and middle turns but remained significantly fewer than controls (**Fig 1H**, **S2F, S2H, and S2M Fig**, **S1 Table**). At P7, P14, and P21, Sox2$^+$ IPhCs of DT-treated $Lgr5^{DTR/+}$ cochleae remained comparable to those in controls, whereas Sox2$^+$ SCs in the lateral compartment degenerated in all 3 turns (**Fig 1H, 1J and 1O**, **S2F, S2H, S2J, S2L and S2M Fig**). Also, we found progressive degeneration of OHCs throughout the DT-treated $Lgr5^{DTR/+}$ cochleae beginning at P14 and P21 (**Fig 1N and 1P**). No degeneration of IHCs was detected (**S2N Fig**). These results indicate spontaneous regeneration of IPhCs and survival of IHCs in the medial compartment and progressive degeneration of SCs and OHCs in the lateral compartment, suggesting a compartmentalized regenerative response to ablation of Lgr5$^+$ SCs.

## Mitotic progenitors in the GER after ablation of Lgr5$^+$ cells

Ablation of HCs in the neonatal cochlea stimulates limited proliferation that is restricted to the apical turn [12,14], whereas SC regeneration was previously reported to be nonmitotic [15]. To investigate whether proliferative regeneration occurs after ablation of Lgr5$^+$ SCs and whether this proliferation is restricted to the apical turn, we administered EdU (25 μg/g, daily P3 to P5, IP) to DT-treated $Lgr5^{DTR/+}$ mice and immunostained for Ki67 (**Fig 2A**). EdU labels actively dividing cells shortly after injection and serves as a mitotic tracer, whereas Ki67 marks nonquiescent cells outside G$_0$ at the time of tissue fixation. As expected, control cochleae showed little to no proliferation throughout the sensory epithelia from P4 to P21, confirming mitotic quiescence (**Fig 2B–2D, 2H–2K and 2O**, **S3A, S3C, S3E, S3G, S3I, S3K, S3N, S3O and S3P Fig**, **S3 and S4 Tables**). In the P4 $Lgr5^{DTR/+}$ damaged cochleae, we noted many EdU$^+$ or Ki67$^+$ Sox2$^+$ SCs in all 3 turns (**Fig 2E–2H**, **S3 and S4 Tables**). In the apex, there were 79.5 ± 17.0 EdU$^+$ and 86.4 ± 20.6 Ki67$^+$ Sox2$^+$ SCs (per 160 μm cochlear length) (**S3 and S4 Tables**). Proliferative cells were present in an apical-to-basal decreasing gradient (base has 29.3 ± 9.4 EdU$^+$ and 36.6 ± 13.3 Ki67$^+$ Sox2$^+$ SCs) and were much more abundant in the GER than organ of Corti (**Fig 2E–2H**, **S3 and S4 Tables**). Specifically, proliferation was the most robust among GER cells adjacent to IPhCs, with fewer Ki67$^+$ and EdU$^+$ cells observed in the lateral compartment (**Fig 2E–2H**, **S3 and S4 Tables**).

Three days later at P7, the number of EdU$^+$ Sox2$^+$ cells in the GER significantly decreased, while EdU$^+$ cells in the IPhC region significantly increased in each cochlear turn (**Fig 2L–2O**, **S4 Table**). As the GER normally degenerates between P7 and P10, these results suggest that

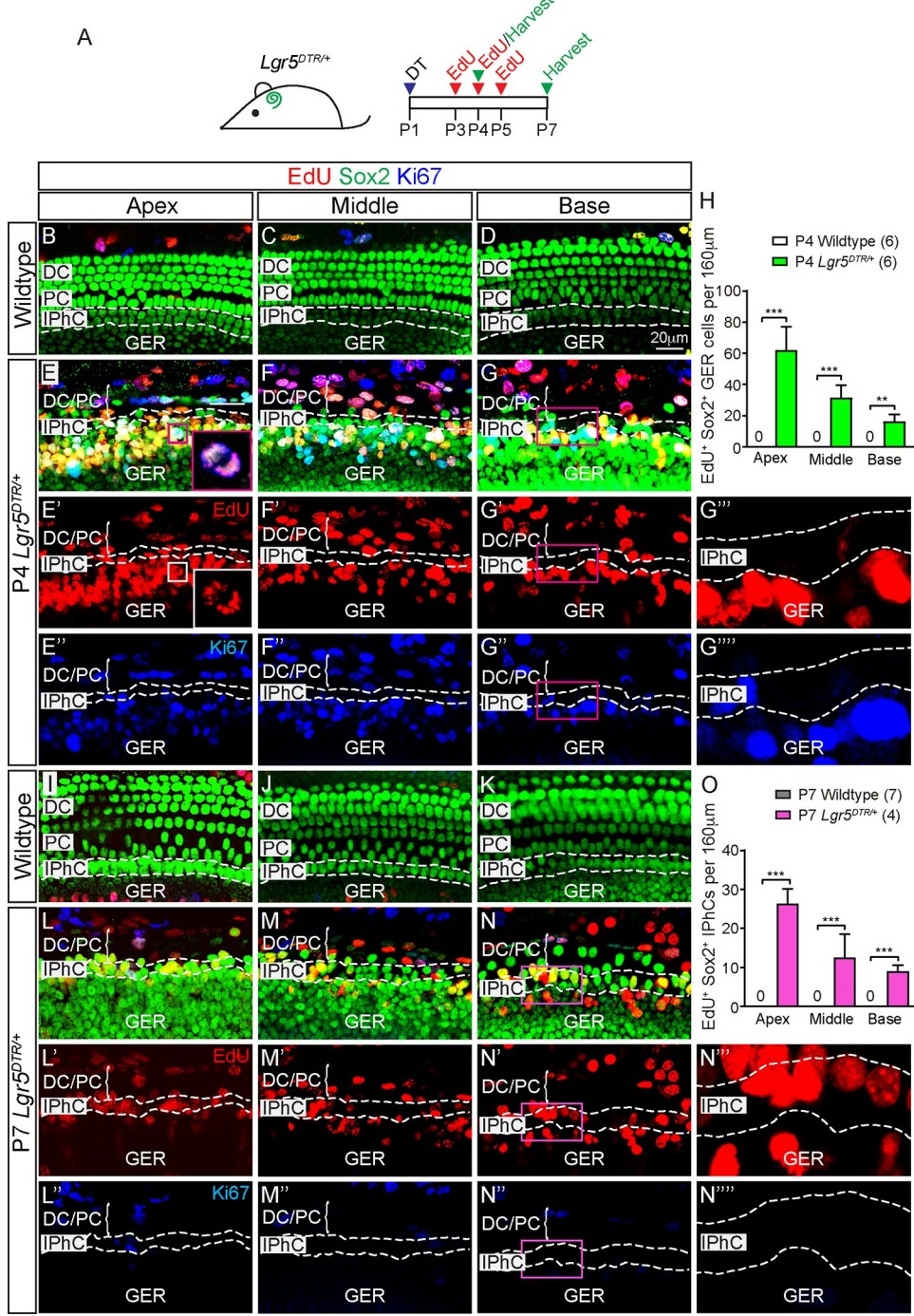

**Fig 2. Mitotic regeneration of IPhCs.** (A) Experimental paradigm: DT was injected into P1 wild-type or *Lgr5^{DTR/+}* mice. EdU was injected daily from P3 to P5, and cochleae were examined at P4 or P7. (B-D, I-K) Representative images of the apical, middle, and basal turns of DT-treated wild-type cochleae showing no EdU⁺ or Ki67⁺ Sox2⁺ SCs at P4 or P7. IPhC region is outlined by dashed lines. (E-G) In each turn of the P4 DT-treated *Lgr5^{DTR/+}* cochlea, EdU⁺ and/or Ki67⁺ Sox2⁺ cells were detected in the GER. Some EdU⁺ and/or Ki67⁺ Sox2⁺ cells were also found in the IPhC and PC/DC regions. E'-G'''represent single channel images. Insets in E and E' are high magnification images of a cell in metaphase. G''' and G''''are high magnification images from G' and G". (H) Significantly more EdU⁺ Sox2⁺ cells were found in the P4 GER regions throughout the DT-treated *Lgr5^{DTR/+}* cochlea relative to controls. Numbers decrease in an apical–basal gradient. (L-N) In all 3 turns of the P7 DT-treated *Lgr5^{DTR/+}* cochlea, EdU⁺ Sox2⁺ cells were primarily found in IPhC region, and only few were found in the GER or PC/DC region. Ki67⁺ Sox2⁺ cells were rarely detected at this age. L'-N''''represent single channel images. N''' and N''''are high magnification images from N' and N". (O)

Significantly more EdU$^+$ Sox2$^+$ cells were detected throughout the P7 IPhC regions of DT-treated *Lgr5$^{DTR/+}$* cochlea relative to controls, decreasing in an apical–basal gradient. Data represent mean ± SD. **$p < 0.01$, ***$p < 0.001$. Two-way ANOVA with Tukey's multiple comparisons test. $n$ = 4–7. See S1 Data for H, O. DC, Deiters' cell; DT, diphtheria toxin; GER, greater epithelial ridge; IPhC, inner phalangeal cell; PC, pillar cell; SC, supporting cell.

some EdU$^+$ GER cells have degenerated, while others migrated to repopulate lost IPhCs. EdU-labeled IPhCs remained present in the P14 and P21 cochlea, demonstrating survival of regenerated cells (S3B, S3D, S3F, S3H, S3J, S3L and S3M Fig, S4 Table). To pinpoint the timing of damage-induced proliferation, we immunostained and found that Ki67$^+$ Sox2$^+$ cells in both the GER and IPhC regions decreased at P7 relative to P4 and were not detected in any regions at P14 or P21 (Fig 2E–2G and 2L–2N, S3B, S3D, S3F, S3H, S3J and S3L Fig, S3 Table). Moreover, when we delayed EdU injection to P7 to P9, we found no EdU-labeled cells in the sensory epithelia (S3Q–S3S Fig). Collectively, these results indicate that SC loss stimulates robust, yet transient proliferation in the GER and damage-activated proliferative cells may migrate laterally to replace lost IPhCs within the first week after insult.

To elucidate the relationship between the degree of SC loss and mitotic regeneration, we injected P1 *Lgr5$^{DTR/+}$* mice with escalating DT doses (0.1, 0.5, 2, and 4 ng/g), followed by EdU at P3 to P4 and harvested cochleae at P4. Increasing DT doses led to greater losses of Sox2$^+$ SCs in both the lateral compartment and the IPhC region (S5 Table). Concomitantly, higher DT doses led to more EdU$^+$ Sox2$^+$ SCs in both the lateral and medial compartments (S6 Table). At high DT doses (2 and 4 ng/g), proliferative cells were consistently found to be more abundant in the medial than lateral compartments throughout the cochlea, and also in an apical–basal gradient (S6 Table). To directly compare these to previous results [15], we examined the *Plp1$^{CreERT/+}$*; *Rosa26R$^{DTA/+}$*; *Rosa26R$^{tdTomato/+}$* (*Plp1-DTA-tdTomato*) mice [25–27], where the *Plp1* lineage represents IPhCs. After tamoxifen administration to control, *Plp1-tdTomato* animals (P0 to P1, 0.075 mg/g, IP), many IPhCs were tdTomato-labeled at P4 (S4A–S4D Fig). Consistent with previous results [15], this tamoxifen regimen to *Plp1-DTA-tdTomato* led to a significant loss of tdTomato$^+$ IPhCs and only rare EdU$^+$ cells in the lateral GER (S4E–S4I Fig). As Lgr5$^+$ cells include both the IPhCs and lateral GER cells, the proliferative response observed in the GER of damaged *Lgr5-DTR* cochleae can be attributed to both the broader and overall more severe depletion of SCs.

## GLAST-Cre$^+$ GER cells mitotically regenerate inner phalangeal cells

To confirm the ablation of IPhCs in *Lgr5-DTR* mice, we labeled for Fabp7, a marker predominantly expressed in IPhCs [24]. Cryosections and whole mounts of P4 and P7 control cochleae showed Fabp7 expression outlining the IPhCs (Fig 3A-A' and 3C-C'). By contrast, Fabp7$^+$ IPhCs were absent after DT-induced damage at P4 and were regenerated by P7, albeit noticeably more disorganized relative to controls (Fig 3B-B' and 3D-D'). At P4, there were significantly fewer Fabp7$^+$ IPhCs in *Lgr5$^{DTR/+}$* cochleae compared to controls (Fig 3E-E', 3F-F' and 3I). P7 *Lgr5$^{DTR/+}$* cochlea display significantly more IPhCs relative to P4, and these numbers were not significantly different from controls (Fig 3G-G', 3H-H', and 3I). Notably, the degree of damage at P4 as well as regeneration of Fabp7$^+$ IPhCs was similar in all 3 turns of the cochlea (Fig 3I).

To test whether proliferative cells in the GER serve as the source of regenerated IPhCs, we fate-mapped and compared the *GLAST* and *Plp1* lineages, which represent cells that reside in the GER and IPhC regions (GLAST-Cre$^+$) and the IPhC region only (Plp1-Cre$^+$) (Fig 3J, S5A Fig). We first performed fate mapping using the *GLAST$^{CreERT/+}$* mice [28]. Tamoxifen was administered at P1 to induce labeling by the Cre reporter line Rosa26R-tdTomato, followed

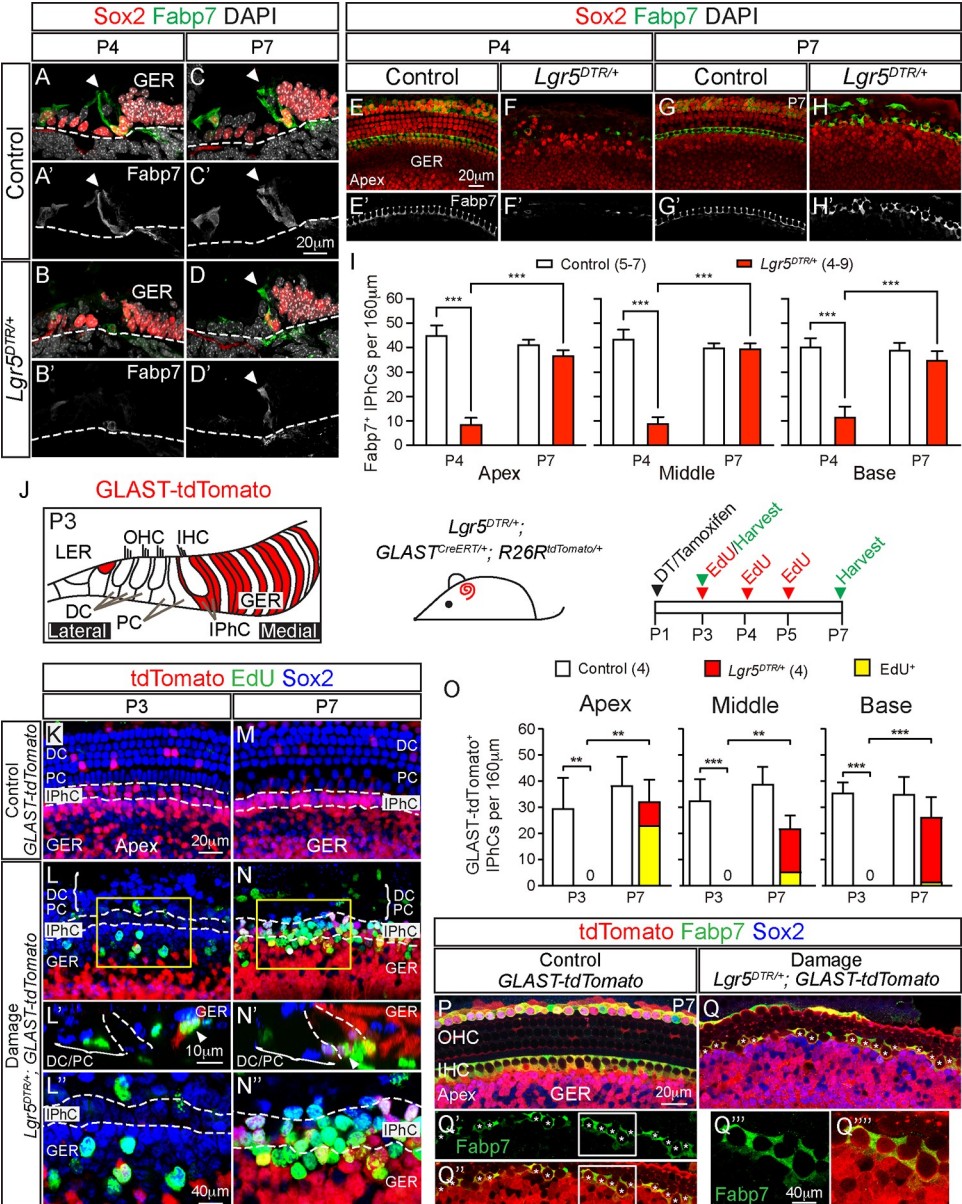

**Fig 3. GLAST-Cre⁺ cells from the GER mitotically regenerate IPhCs.** Representative images of apical turns are shown. (A, C) Cryosections of P4 and P7 control cochleae with Fabp7⁺ Sox2⁺ IPhCs (arrowheads). (B) Most Fabp7⁺ Sox2⁺ IPhCs have degenerated in P4 DT-treated *Lgr5^{DTR/+}* cochlea. (D) In the P7 DT-treated *Lgr5^{DTR/+}* cochlea, many Fabp7⁺ Sox2⁺ IPhCs were found. Sox2⁺ SCs in the lateral compartment have degenerated at this age. (E, G) Merged z-stack images of whole mount preparation of control cochleae at P4 and P7 showing Fabp7⁺ Sox2⁺ IPhCs. (F) Most Fabp7⁺ Sox2⁺ IPhCs have degenerated in the P4 DT-treated *Lgr5^{DTR/+}* cochlea. (H) Replenished Fabp7⁺ Sox2⁺ IPhCs in P7 DT-treated *Lgr5^{DTR/+}* cochlea. (E'-H') Single-plane images from the same representative sample shown in (E-H) highlighting Fabp7⁺ IPhCs. (I) Quantification showing significantly fewer Fabp7⁺ IPhCs in DT-treated *Lgr5^{DTR/+}* cochlea relative to controls and significantly more Fabp7⁺ IPhCs in all cochlear turns at P7 than P4. (J) Schematic of GLAST-tdTomato expression in the P3 *GLAST^{CreERT/+}; R26R^{tdTomato/+}* cochlea (tamoxifen at P1). Experimental paradigm using *GLAST^{CreERT/+}; R26R^{tdTomato/+}* (control) or *Lgr5^{DTR/+}; GLAST^{CreERT/+}; R26R^{tdTomato/+}* mice (damage). DT and tamoxifen were injected at P1, EdU from P3 to P5, and cochleae examined at P3 and P7. (K, M) In the P3 and P7 control cochlea, there were no EdU⁺ tdTomato⁺ Sox2⁺ SCs in the GER. Some Sox2⁺ IPhCs were tdTomato⁺, and none were EdU⁺. (L, N) In the P3 damage cochlea, EdU⁺ tdTomato⁺ Sox2⁺ SCs were detected in the GER but not in the IPhC region. At P3, many Sox2⁺ IPhCs had degenerated, and none were tdTomato⁺. At P7, many EdU⁺ tdTomato⁺ Sox2⁺ SCs occupied both the GER and the IPhC regions. Orthogonal images (L' and N') showing EdU⁺ tdTomato⁺ Sox2⁺ cells in the GER at P4 and in the IPhC region at P7. L" and N" are high magnification images from L and N. (O) Quantification of (total and EdU⁺) GLAST-tdTomato⁺ Sox2⁺ IPhCs at P3 and P7. Control cochleae had no

EdU⁺ GLAST-tdTomato⁺ Sox2⁺ IPhCs at both ages in all turns examined. In the damaged cochleae, there were significantly more tdTomato⁺ Sox2⁺ IPhCs (both EdU⁺ and EdU-negative) at P7 relative to P3. Though present in all 3 turns, EdU⁺ GLAST-tdTomato⁺ Sox2⁺ IPhCs decreased in an apex-to-base gradient at P7. (P) P7 control cochlea with Fabp7⁺ tdTomato⁺ IPhCs. (Q) P7 damage cochlea with many Fabp7⁺ tdTomato⁺ IPhCs (asterisks) arranged in a disarrayed pattern. Q' and Q" are single and dual channel images of Q. Q'" and Q""are high magnification images of traced Fabp7⁺ IPhCs from Q' and Q". Data represent mean ± SD. **$p < 0.01$, ***$p < 0.001$. Two-way ANOVA with Tukey's multiple comparisons test. $n$ = 4–9. See S1 Data for I, O. DC, Deiters' cell; DT, diphtheria toxin; GER, greater epithelial ridge; IHC, inner hair cell; IPhC, inner phalangeal cell; OHC, outer hair cell; PC, pillar cell; SC, supporting cell.

by EdU injections (daily from P3 to P5) to label proliferating cells (Fig 3J). In the undamaged, $GLAST^{CreERT/+}$; $Rosa26R^{tdTomato/+}$ (GLAST-tdTomato) cochleae, tdTomato⁺ cells occupied the IPhC and GER regions at P3 to P7 (46% to 54% GER cells labeled; Fig 3K and 3M, S7 Table). The expression pattern of GLAST-tdTomato is consistent with previous reports and contrasts with that of Lgr5-DTR-EGFP (Fig 1A). In the P3 DT-damaged $GLAST^{CreERT/+}$; $Lgr5^{DTR/+}$; $Rosa26R^{tdTomato/+}$ (GLAST-tdTomato-DTR) cochlea, there was a loss of tdTomato⁺ IPhCs and emergence of tdTomato⁺, EdU⁺ cells in the GER (Fig 3L-L' and 3O). At P7, many tdTomato⁺ cells have migrated to the IPhC region adjacent to IHCs along the length of the cochlea (Fig 3N-N' and 3O). Remarkably, EdU⁺, tdTomato⁺, IPhCs were observed in all 3 turns of GLAST-tdTomato-DTR cochleae and most numerous in the apex (Fig 3O, S8 Table). Finally, anti-Fabp7 was used to confirm the IPhC identity of fate-mapped cells during regeneration. As expected, GLAST-tdTomato⁺ cells in the IPhC region were Fabp7⁺ in the P7 control cochleae (Fig 3P). In the damaged P4 cochleae, almost all Fabp7⁺ cells were ablated (Fig 3B and 3F), whereas many Fabp7⁺ tdTomato⁺ cells occupied the IPhC region at P7 (Fig 3Q-Q""), indicating differentiation of GER-derived cells into IPhCs. Together, these results suggest that GLAST-Cre⁺ GER cells in the neonatal cochlea mitotically regenerated IPhCs.

To rule out the possibility that surviving IPhCs contributed to the regeneration of IPhCs, we used $Plp1^{CreERT/+}$ mice for lineage tracing [25] (S5A and S5B Fig). In the P3 and P7 undamaged $Plp1^{CreERT/+}$; $Rosa26R^{tdTomato/+}$ (Plp1-tdTomato) cochleae, IPhCs were selectively labeled with tdTomato as before (S5C–S5E Fig). In the DT-treated $Plp1^{CreERT/+}$; $Rosa26R^{tdTomato/+}$; $Lgr5^{DTR/+}$ cochlea (Plp1-tdTomato-DTR), there were almost no tdTomato⁺ IPhCs at any stage examined, while EdU⁺ GER cells first started to appear at P3 and later occupied the IPhC region at P7 (S5F–S5K Fig, S7 and S8 Tables), suggesting that IPhCs do not self-regenerate.

## Regenerated cells display cochlear supporting cell properties

We next sought to characterize regenerated IPhCs in the P7 $Lgr5^{DTR/+}$ cochlea, a time point when IPhC counts had returned to control levels. In the neonatal cochlea, waves of ATP-mediated spontaneous calcium activity are deemed critical for cochlear maturation also spread through this network of SCs, particularly those in the IPhC and GER regions [8]. We first determined whether regenerated IPhCs display spontaneous Ca²⁺ transients, which could imply differentiation and integration of regenerated SCs [7,8]. We generated the transgenic mouse strain $Lgr5^{DTR/+}$; $Atoh1$-mCherry; $Pax2$-Cre; $Rosa26R^{GCaMP3/+}$ (Fig 4A) [7,29], allowing live imaging of HCs (Atoh1-mCherry) and spontaneous calcium transients ($Pax2$-Cre; $Rosa26R^{GCaMP3/+}$) after ablation of Lgr5⁺ cells ($Lgr5^{DTR/+}$).

In the undamaged, control cochleae, spontaneous calcium transients were periodically detected in the lateral GER and IPhC regions (Fig 4B, 4D and 4F–4H, S1 Movie), with frequency, size, and intensity consistent with previous results [7]. In the damaged cochlea, spontaneous calcium transients spanning the IPhC and the GER regions appeared more frequently

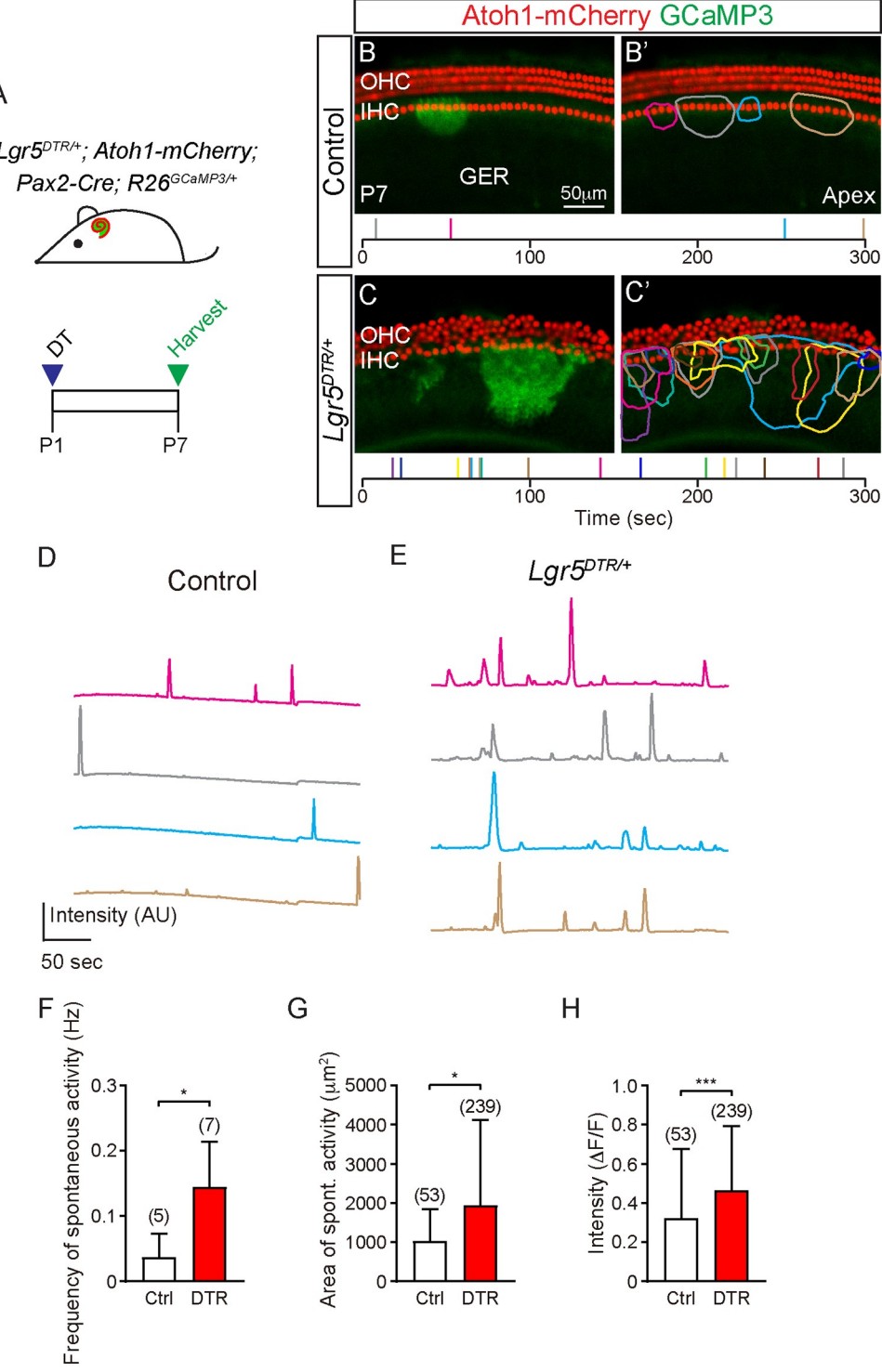

**Fig 4. Physiological properties of regenerated cochlea.** (A) Experimental paradigm for live imaging: DT was injected into *Lgr5*$^{DTR/+}$; *Atoh1-mCherry*; *Pax2-Cre*; *R26*$^{GCaMP3/+}$ and *Atoh1-mCherry*; *Pax2-Cre*; *R26*$^{GCaMP3/+}$ mice at P1 and cochleae were examined at P7. (B–C') Representative still images from the apical turn are shown (see S1 Video). Without damage, periodic calcium (EGFP) signals (ROI outlined in colors that correspond to the bars in the time line plotted) were detected in lateral GER cells and IPhCs surrounding the Atoh1-mCherry$^+$ IHCs. In the damaged cochlea, more EGFP$^+$ events were detected adjacent to IHCs, with each event spanning a wider area and appearing more intense when compared to controls. (D, E) Representative tracings of EGFP signals measured from individual ROIs in undamaged and damaged organs. Colors correspond to those in B'–C'. Tracing from 4 events from C' were

shown in E. (F-H) Frequency, areas, and relative intensity change of spontaneous calcium activities in damaged cochleae were significantly greater than those in controls. Data represent mean ± SD. $^*p < 0.05$, $^{**}p < 0.01$, $^{***}p < 0.001$. Unpaired Student $t$ test or Mann–Whitney test. $n$ = 5–239 cells from 5–7 cochleae. See S1 Data for F-H. DT, diphtheria toxin; GER, greater epithelial ridge; IHC, inner hair cell; IPhC, inner phalangeal cell; OHC, outer hair cell; ROI, region of interest.

(**Fig 4C and 4E**, **S1 Movie**). Additionally, spontaneous calcium transients in the damaged cochleae were more intense and spanned larger areas relative to controls (**Fig 4D–4H**). These properties are reminiscent of SCs in the perinatal cochlea [30] and suggest the presence of an SC network incorporating the newly regenerated cells. Taken together, these results indicate that regenerated SCs exhibit characteristics of neonatal IPhCs and may be connected to native SCs.

## Compartmentalized regeneration, cell survival, and maturation

Regenerated HCs in the neonatal cochlea are short-lived and undergo delayed degeneration [12,14]. In the lateral compartment, CD44 marks outer pillar cells and the lesser epithelial ridge (LER) cells, and Sox2 labels DCs and pillar cells (**S6A Fig**) [20,31]. The lateral compartment of P7 DT-treated $Lgr5^{DTR/+}$ cochlea displayed a loss of Sox2$^+$ and CD44$^+$ SCs and abnormally clustered Myosin7a$^+$ outer HCs (**S6B Fig**), confirming degeneration without significant regeneration. These results are consistent with previous results when DCs and pillar cells in the lateral compartment were ablated [32].

On the other hand, regenerated IPhCs in the medial compartment have been shown to survive in the mature cochlea [15]. To assess whether regenerated IPhCs in the $Lgr5^{DTR/+}$ mice survive and undergo maturation, we examined P14 and P21 cochleae. At these ages, IHCs have matured to express Vglut3, and IPhCs have begun to express the mature markers Na$^+$/K$^+$ ATPase α-1 and GLAST (**Fig 5A, 5C, 5E and 5G**, **S6C and S6E Fig**) [33–35]. By P14, the GER in control cochlea has undergone apoptosis leading to the formation of the inner sulcus. In the lateral compartment of regenerated $Lgr5^{DTR/+}$ cochlea, SCs transiently and modestly increased between P4 and P7, before progressively degenerated between P7 to P21 (**S2M Fig**, **S1 Table**). Though present at P7, OHCs also degenerated at P14 and onwards (**Figs 1M**, **1N**, **1P**, **5B and 5D**, **S1 Table**). Thus, regeneration in the lateral compartment is limited, and cells undergo progressive degeneration after ablation of Lgr5$^+$ SCs. In stark contrast, robust regeneration and survival of regenerated cells were observed in the medial compartment of $Lgr5^{DTR/+}$ cochleae. At P14 and P21, there was no detectable loss of IHCs or IPhCs, which matured to express Vglut3 and Na$^+$/K$^+$ ATPase α-1/GLAST, respectively (**Fig 5F and 5H**, **S6D and S6F Fig**, **S9 Table**). The numbers of IHCs and IPhCs in $Lgr5^{DTR/+}$ cochleae remained comparable to controls at P14 and P21 (Fig **5S**, **S2N Fig**, **S9 Table**). At P14 and P21, mitotically regenerated tdTomato$^+$ IPhCs matured to express GLAST protein in the all 3 turns, indicating differentiation and long-term survival (**Fig 5I, 5J–5J', and 5K–5L**, **S6G and S6H–S6H' Fig**, **S7 and S8 Tables**). These results suggest that regenerated IPhCs are at least partially mature and remained viable in the mature cochlea.

Degeneration of HCs during the neonatal period causes retraction of the innervating fibers [36]. In the lateral compartment, Tuj1$^+$ fibers project laterally to innervate OHCs (**Fig 5M**). In the damaged $Lgr5^{DTR/+}$ cochlea, no Tuj1$^+$ fibers were detected in the lateral compartment, likely as a result of OHC degeneration (**Fig 5N**). However, in the medial compartment of DT-treated animals, there was no detectable loss of Vglut3$^+$ IHCs (**Fig 5S**, **S9 Table**), which appeared to remain innervated and juxtaposed to the inner spiral plexus with radial fibers projecting from the spiral ganglia neurons (**Fig 5M–5P**). While the plexus was somewhat

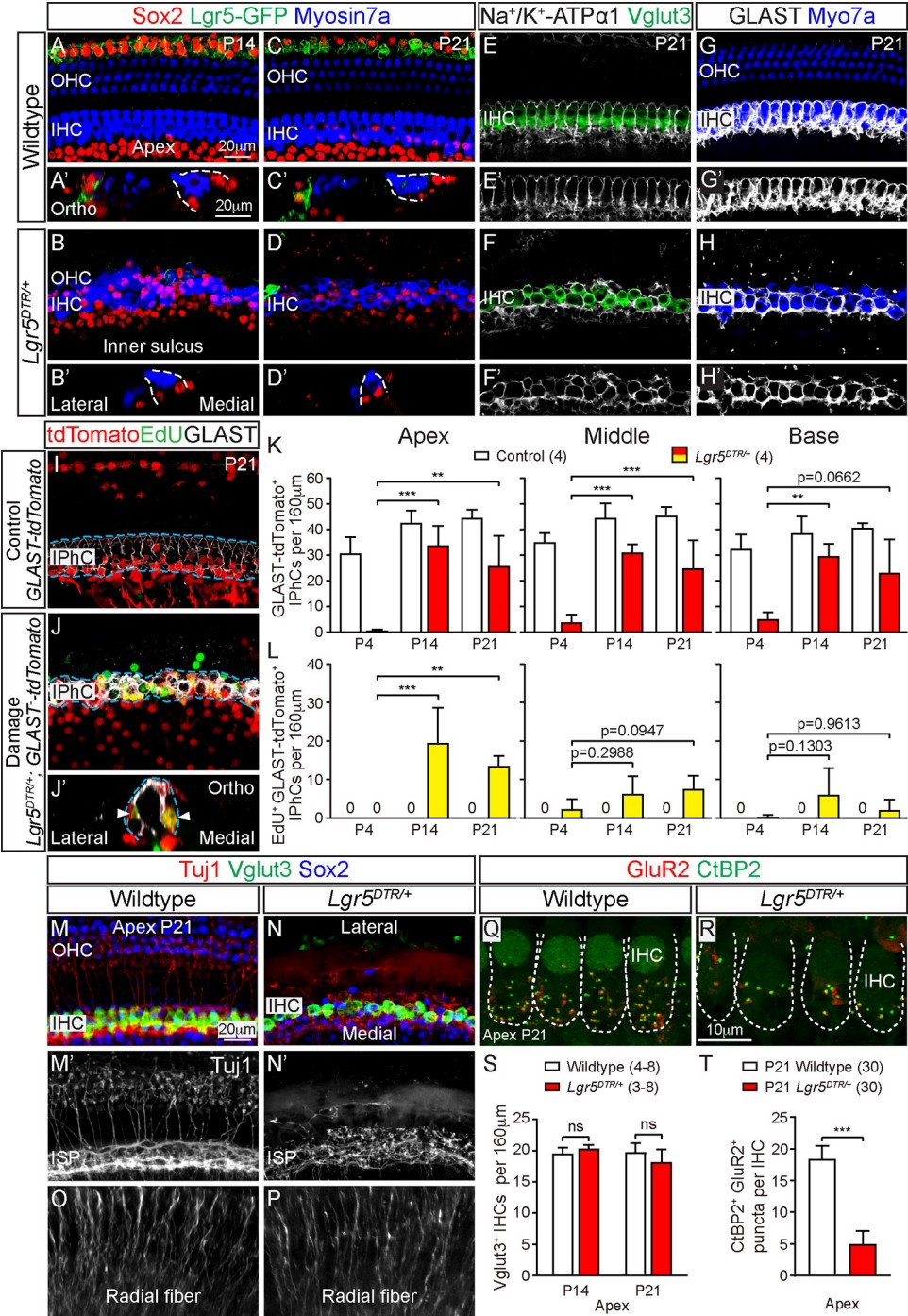

**Fig 5. Maturation of regenerated IPhCs.** (A-D) Representative images of the apical turn are shown. In the P14 and P21 DT-treated *Lgr5^{DTR/+}* cochleae, OHCs have degenerated with a loss of tunnel of Corti, while IHCs and a few Lgr5-EGFP⁺ cells remained. A'-D' show respective orthogonal images. (E-F) In the undamaged (control) P21 cochlea, Vglut3⁺ IHCs were surrounded by Na⁺/K⁺ ATPase α-1⁺ IPhCs. In the DT-damaged *Lgr5^{DTR/+}* cochlea, regenerated Na⁺/K⁺ ATPase α-1⁺ IPhCs and Vglut3⁺ IHCs remained present. E'-H' show respective single-channel images of IPhCs. (G) In the undamaged P21 cochlea, IHCs were surrounded by GLAST⁺ IPhCs. (H) In the P21 DT-treated *Lgr5^{DTR/+}* cochlea, regenerated IPhCs surrounding IHCs expressed GLAST. (I) In the P21 control cochlea, there were no EdU⁺ tdTomato⁺ SCs. GLAST⁺ IPhCs were tdTomato⁺, while none were EdU⁺. (J) In the P21 damaged cochlea, there were many EdU⁺ tdTomato⁺ GLAST⁺ IPhCs. Orthogonal views of EdU⁺ tdTomato⁺ GLAST⁺ IPhCs (arrowheads) shown in J'. (K-L) Quantification of GLAST-tdTomato⁺ GLAST⁺ IPhCs and EdU⁺ GLAST-tdTomato⁺ GLAST⁺ IPhCs in the control and damaged cochleae. None of the GLAST-tdTomato⁺ GLAST⁺ IPhCs were labeled by

EdU$^+$ in the control cochleae at any ages examined. In the regenerated P14 and P21 cochleae, GLAST-tdTomato$^+$ GLAST$^+$ or EdU$^+$ GLAST-tdTomato$^+$ GLAST$^+$ IPhCs were present in all 3 turns. (M) P21 undamaged cochleae with Tuj1$^+$ fibers in the PC/DC region and a dense inner spiral plexus (ISP) adjacent to Vglut3$^+$ IHCs. (N) DT-damaged P21 *Lgr5$^{DTR/+}$* cochlea showing a loss of Tuj1$^+$ fibers in the PC/DC region and a disorganized ISP. (O, P) Tuj1$^+$ radial fibers in P21 control and damaged cochleae. (Q) In the undamaged P21 (control) cochleae, many CtBP2$^+$ GluR2$^+$ synapses were detected in IHCs. (R) In the P21 DT-treated *Lgr5$^{DTR/+}$* cochlea, CtBP2$^+$ GluR2$^+$ synapses were notably reduced. (S) The number of Vglut3$^+$ IHCs in the apical turn of cochleae from DT-treated wild-type control and DT-treated *Lgr5$^{DTR/+}$* mice is comparable at P14 and P21. (T) Quantification showing significantly fewer CtBP2$^+$ GluR2$^+$ puncta in IHCs from P21 damaged cochleae than control. Data represent mean ± SD. $^{**}p < 0.01$, $^{***}p < 0.001$. Two-way ANOVA with Tukey's multiple comparisons test or unpaired Student *t* test. *n* = 3–30. See S1 Data for K, L, S, and T. DC, Deiters' cell; DT, diphtheria toxin; IHC, inner hair cell; IPhC, inner phalangeal cell; IS, inner sulcus; ISP, inner spiral plexus; OHC, outer hair cell; PC, pillar cell; SC, supporting cell.

disarrayed, radial fibers appeared organized and dense, comparable to controls (**Fig 5M–5P**). To probe whether synapses remained present, we examined the presynaptic and postsynaptic markers CtBP2 and GluR2 (**Fig 5Q**) [37]. In the P21 DT-treated *Lgr5-DTR* cochleae, IHCs showed fewer CtBP2$^+$ GluR2$^+$ synapses relative to controls (**Fig 5Q, 5R and 5T**), suggesting synaptopathy despite cell survival. Consistent with these histological findings, auditory brainstem responses (ABRs) and distortion product otoacoustic emissions (DPOAEs) were absent across all frequencies in P21 DT-treated *Lgr5$^{DTR/+}$* mice (**S6I and S6J Fig**). Collectively, these results indicate that ablation of Lgr5$^+$ SCs caused limited regeneration and progressive degeneration in the lateral compartment, while inducing robust regeneration and survival of IPhCs in the medial compartment of the mature cochlea. In addition, ablation of Lgr5$^+$ SCs caused synapse loss in IHCs and loss of OHCs, resulting in profound hearing loss.

## Translatomic analysis of GLAST-Cre$^+$ cells

To begin to determine the molecular mechanisms directing mitotic regeneration, we first analyzed the arrays of genes expressed by undamaged and damage-activated GLAST-Cre$^+$ cells using the transgenic RiboTag mouse [38,39]. This allele permits the immunoprecipitation (IP) of ribosomes from Cre$^+$ cells, enriching for cell type–specific actively translated mRNAs, or translatomes. We generated *GLAST$^{CreERT/+}$; Rpl22$^{HA/+}$* (GLAST-RiboTag, from hereon "control") and *Lgr5$^{DTR/+}$; GLAST$^{CreERT/+}$; Rpl22$^{HA/+}$* (GLAST-RiboTag-DTR, from hereon "DTR") for experiments. Control and DTR mice were injected with DT and tamoxifen at P1, and cochleae were harvested at P4, a time point when damaged-induced proliferation was robust (**Fig 6A–6C** ). RNA was extracted both from whole sensory epithelia (input) as well as ribosome immunoprecipitated samples (IP) to enrich for translatomes of GLAST-Cre$^+$ cells and allow calculation of an enrichment factor (EF) for each gene [40,41]. This index was then used to gauge whether signal originates in the cell type immunoprecipitated (EF > 2) or primarily from other cell types.

Unsupervised hierarchical clustering analysis identified striking translatomic differences between sample types (input versus IP) and treatment groups (control versus DTR), which, in a principal component analysis (PCA), correspond to PC1 and PC2, respectively, while genetic convergence among the biological replicates was seen within each experimental group (**Fig 6D and 6E, S10 Table**).

From GLAST-Cre$^+$ cells in the control cochleae, we identified 147 genes that were both highly enriched in the IP compared with the input (LFC ≥ 2) and had an expression value greater than the median counts per kilobase million (CPM) value (**S11 Table**). All enriched ontological biological processes associated with these genes were related to transcription regulation (38 genes, FDR 7.13E-11) (**S10 Table**). Among these genes, we validated 5 members (*Adrb2*, *Igf2bp1*, *Igf2*, *Socs1*, and *Socs3*) and also 4 additional enriched genes (*Igfbp3*, *Igf2bp3*,

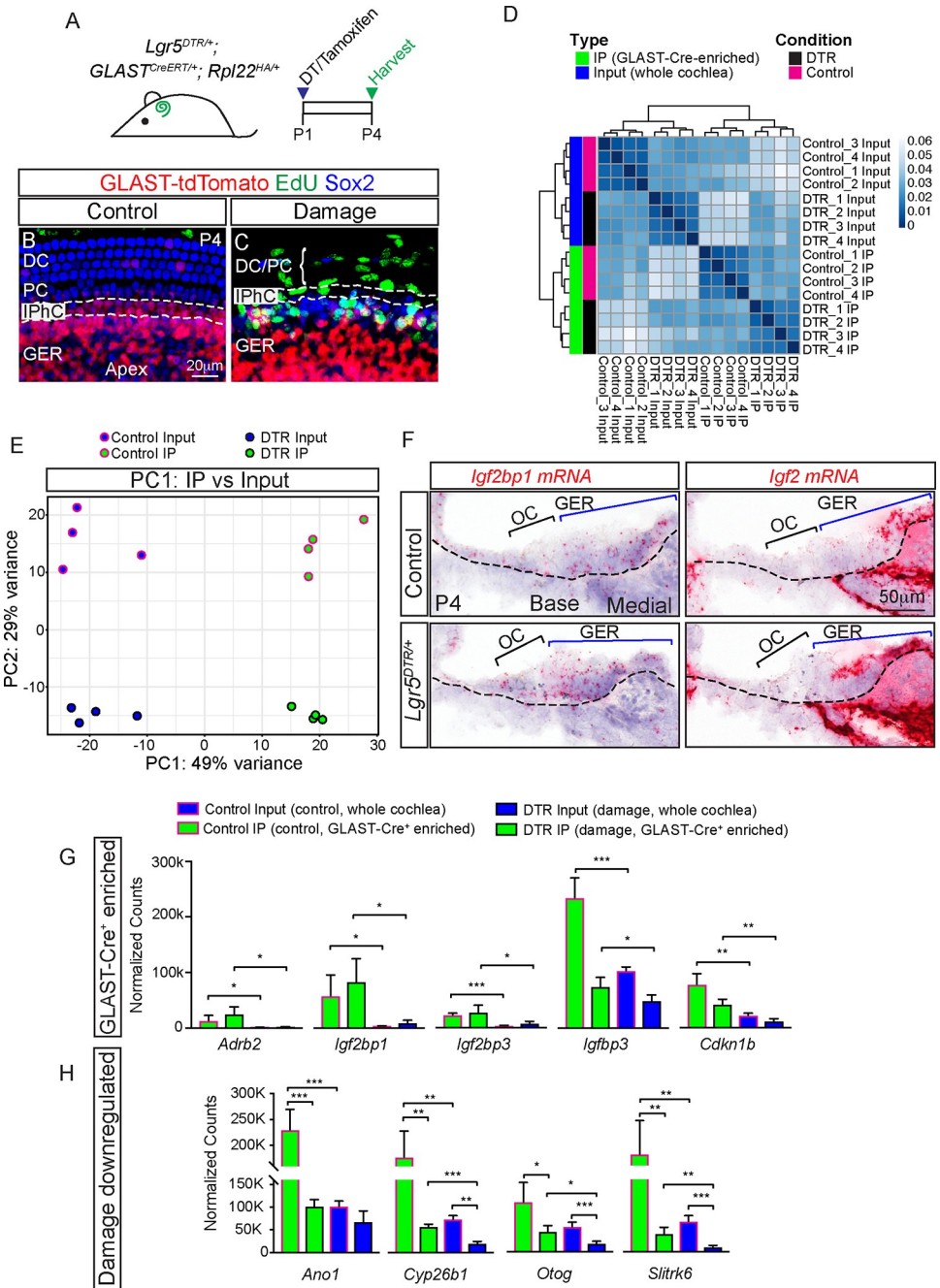

**Fig 6. Translatomic analysis of GLAST-Cre+ cells.** (A) Schematic displaying the mouse model and experimental timeline of DT and tamoxifen administration at P1 followed by cochlea harvest from P4 *Lgr5DTR/+*; *GLASTCreERT/+*; *Rpl22HA/+*. (B) GLAST-tdTomato is detected in IPhCs and GER cells in P4 *GLASTCreERT/+*; *R26RtdTomato/+* (control) cochleae. (C) EdU+ tdTomato+ Sox2+ cells are found in the GER of the DT-treated *Lgr5DTR/+*; *GLASTCreERT/+*; *R26RtdTomato/+* (DTR) cochleae. Dashed lines highlight IPhC region. (D) Four groups were analyzed: control and damage input (whole cochlea), control and damage immunoprecipate (IP, GLAST-Cre+ cells) with unsupervised hierarchical clustering analysis between samples showing higher correlation between biological replicates within each experimental group than with samples of other experimental groups. (E) PCA. Samples were projected onto the first 2 principal components, which were identified as cellular enrichment (IP vs. Input) as PC1 and the status of the cells (control vs. DTR) as PC2. (F) RNAscope in situ hybridization was used to validate a subset of differentially expressed and enriched genes, 2 of these genes *Igf2bp1* and *Igf2* were spatially enriched in the GER in the P4 control and damage cochlea (basal turn shown). (G, H) Validation of 5 enrichment and 4 damage down-regulated genes, respectively, using nCounter. Data represent mean ± SD. $^*p < 0.05$, $^{**}p < 0.01$, $^{***}p < 0.001$ (Student *t* test). *n* = 4. See S1 Data for G and

H. DC, Deiters' cell; DT, diphtheria toxin; DTR, diphtheria toxin receptor; GER, greater epithelial ridge; IP, immunoprecipitation; IPhC, inner phalangeal cell; PC, pillar cell; PCA, principal component analysis; OC, organ of Corti.

*Cdkn1b*, and *Dlx5*) using nCounter (**Fig 6G, S7A Fig**). In addition, we performed *in situ* hybridization and found *Igf2* and *Igf2pb1* to be spatially enriched in the control and damaged GER, thereby confirming that GLAST-Cre+ cells are representative of GER cells (**Fig 6F, S7B–S7D Fig**). While our data on *Igf2* and *Igfbp3* are consistent with previous results [42], other genes have not been characterized in the cochlea and therefore serve as novel markers of the GER. These data indicate that GLAST-Cre+ GER cells have unique molecular signatures in both the undamaged and damaged cochleae.

To delineate damage-induced changes in gene expression, we compared control IP and DTR IP samples and identified 603 DEGs (LFC ≥ 1, FDR < 0.05, separation > 1.5) (**S12 Table**). These genes were enriched for biological processes highly associated with cell division (e.g., mitotic nuclear division, $p$ = 2.28E-17) (**S10 Table**), consistent with the observed damaged-induced proliferation and increased *Mki67* mRNA levels (**S8A Fig**).

We validated 24 DEGs between control and damaged-activated GER cells using nCounter, including 7 damage-down-regulated (*Ano1*, *Cyp26b1*, *Otog*, *Slitrk6*, *Lfng*, *Ppp2r2b*, and *Trh)* and 8 damage-up-regulated genes (*Egr1*, *Egr4*, *Atf3*, *Iqck*, *Cacna1*, *Ccnb2*, *Cdk1*, and *Cenpf)* (**Figs 6H, 7C, 7D, and 7G–7J, S8C and S8D Fig, S12 Table**). Of these, *Ccnb2*, *Cdk1*, and *Cenpf* are known regulators of cell cycle, which again support the observed damaged-induced proliferation in GER cells.

## Distinct patterns of gene expression

To further classify patterns of differential gene expression across sample types and treatment groups, unbiased cluster analysis was performed using the CLICK algorithm within the Expander gene expression platform [43]. Using this technique, we identified 19 main clusters within the 5,095 DEGs from both control and damaged input and IP groups, 11 of which contained more than 40 genes (**Fig 7A–7K, S13 Table**). We classified these 11 clusters into 5 main categories: (1) Glast-Cre+ enriched (cluster 1); (2) Glast-Cre+ depleted (cluster 2); (3) damage down-regulated (clusters 5, 6, 7, 10); (4) damage up-regulated (clusters 3, 4, 8, 9); and (5) differentially regulated between input and DTR (cluster 11).

Cluster 1 includes *Dlx5*, *Socs1*, *Socs3*, *Adrb2*, and *Igf2*, which represent newly validated and previously reported markers of GER in the medial domain (**Figs 6G, 6H and 7A**) [42]. Cluster 2 primarily comprises markers of HCs (*Fgf8*, *Myo6*, *Kcna10*, *Pvalb*, and *Efemp1*) (**Fig 7B**). A wide array of genes contribute to the damage down-regulated clusters (clusters 5, 6, 7, and 10), including markers of SCs and GER (*Lfng*, *Igfbp3*) and thus reflective of cell loss (**Fig 7C–7F**) [44], along with those previously validated (*Ano1*, *Cyp26b1*, *Slitrk6*; **Fig 6H**). Additionally, other cohorts of genes that decreased include modulators of myelination (*Mpb*, *Pou3f1*) [45,46], innervation (*Slitrk6*) [47], cell adhesion (*Nrcam*), and gap junction genes (*Gjb1*) (**Figs 6H and 7C–7F, S13 Table**) [47,48]. Interestingly, although OHCs remained in the P4 damaged cochlea (**Fig 1L, S1 Table**), the OHC gene *Kcnq4* was down-regulated, while *Ocm* and *Slc26A5* remained unchanged (**Fig 7F, S13 Table**). *Ano1*, which is expressed by IPhCs and GER and required for spontaneously calcium activity [7], was also down-regulated (**Figs 6H and 7C**). Consistent with robust proliferation after damage, the cell cycle inhibitor *Cdkn1b* (encoding p27kip1) was down-regulated (**Figs 6G and 7D**) [49], while other known and candidate regulators of proliferation were up-regulated after damage (clusters 3, 4, 8, and 9), including *Cdk1*, *Cenpf*, *Ccar2*, *Cdc20*, *Cdca2*, *Cdca3*, and *E2f5* (**Fig 7G–7J, S13 Table**). *Egr1*, a

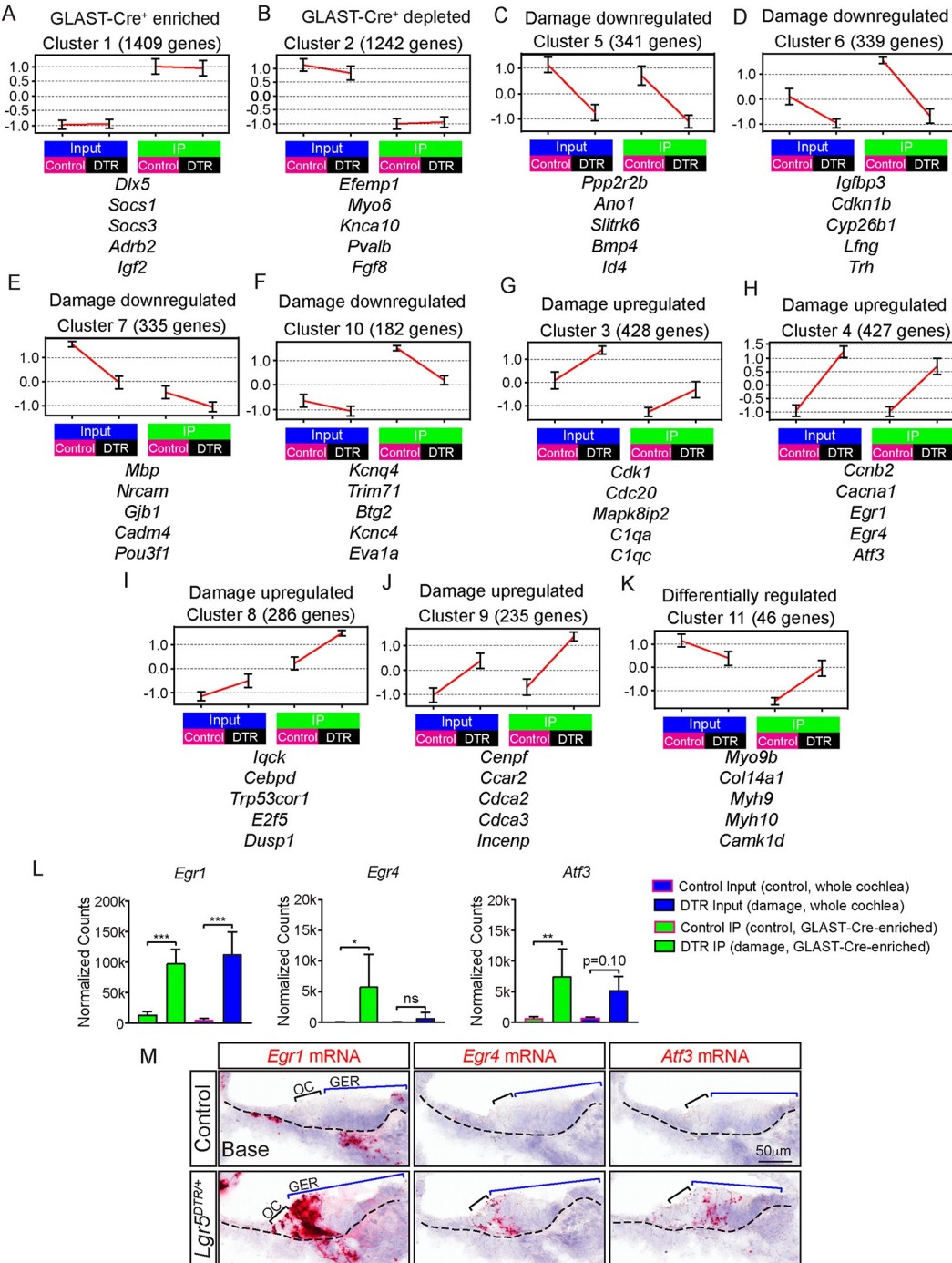

**Fig 7. Analysis of gene clusters of the damaged GER.** (A-K). Cluster analysis applied to the set of 5,095 DEGs, which identified 11 main expression patterns grouped into 5 categories. (A) genes enriched in GLAST-Cre[+] population, (B) genes depleted in the GLAST-Cre[+] population, (C-F) genes expressed at lower levels after damage, (G-J) genes expressed at higher levels after damage, and (K) genes differentially regulated between input and IP after damage. Each cluster is represented by its mean expression patterns (error bars represent ±SD). A biologically representative group of genes is displayed under each cluster. Expression levels were standardized per gene prior to clustering in order to group genes by their expression pattern, regardless of absolute expression magnitude. (L) Normalized counts using the nCounter assay showed that *Egr1*, *Egr4*, and *Atf3* mRNA expression were significantly increased in damaged (DTR) IP samples compared to undamaged (control) IP samples. (M) At P4 *Egr1*, *Egr4*, and *Atf3*, mRNA expression was minimal in the undamaged cochleae, becoming robust in the OC and lateral GER after damage. Data represent mean ± SD. *$p < 0.05$, **$p < 0.01$, ***$p < 0.001$. One-way ANOVA with Tukey's multiple comparisons test, Mann–Whitney test, or Student *t* test. *n* = 4–5. See S1 Data for A-L. DC, Deiters' cell;

DEG, differentially expressed gene; DTR, diphtheria toxin receptor; GER, greater epithelial ridge; IHC, inner hair cell; IP, immunoprecipitation; OHC, outer hair cell; IPhC, inner phalangeal cell; OC, organ of Corti; PC, pillar cell.

transcription factor implicated in liver regeneration [50], along with its associated genes (*Egr4* and *Atf3*), were also dramatically up-regulated after damage (cluster 4) (**Fig 7H**). *In situ* hybridization and nCounter (see Methods for details) both confirmed minimal *Egr1*, *Egr4*, and *Atf3* mRNA expression in the undamaged cochlea and their robust increase after damage (**Fig 7H**, **7L and 7M**, **S8E–S8G Fig**). Finally, genes differentially regulated between input (decrease) and IP samples (increase) after damage included those that regulate actin filament–based movement (*Myo9b*, *Myh10*, *Myh9*) (cluster 11), which may play a role of cell migration (**Fig 7K**) [51–53]. Collectively, these results demonstrate enriched GER genes and damage-up-regulated and down-regulated genes, which may drive proliferation and/or migration of mitotic SC progenitors in the GER.

## Discussion

Cellular plasticity in several mammalian organs, such as the adult liver, pancreas, and brain, facilitates regeneration in response to injury [54]. While the mature cochlea lacks the ability to regenerate, the neonatal mouse cochlea harbors Lgr5$^+$ progenitors that act to replace lost HCs [12,14] and can serve as a renewable source of sensory cells [55,56]. Previously, using multiple CreER lines to express DTA, Mellado Lagarde and colleagues reported that the GER cells nonmitotically regenerate the SC subtype IPhCs [15]. Here, through the use of fate mapping and EdU pulse-chase experiments in the *Lgr5-DTR* mouse line, we reveal that loss of Lgr5$^+$ SCs activates GER cells to proliferate and migrate into the organ of Corti to replace the lost IPhCs. Notably, these newly regenerated cells mature to display features of differentiated SCs (i.e., expression of Glast, Na$^+$/K$^+$ ATPase α-1, and Fabp7) and survive in the mature cochlea and likely represent the source of SC regeneration noted previously [15]. Lastly, we have unveiled the translatomes of quiescent and damage-activated GER cells and candidate genes, which may regulate mitotic regeneration in the neonatal cochlea.

### Discovery of a hidden progenitor cell population in the cochlea

Cochlear SCs also serve the critical function of maintaining ionic homeostasis, mediating purinergic signaling required for spontaneous calcium activity, and supplying neurotrophic support essential for afferent innervation [5–8]. Despite the multiple roles of SCs, the mechanism by which SCs are regenerated is largely unknown.

The GER has long been known as a transient structure located adjacent to the organ of Corti; however, its function remains incompletely understood [17,57]. Previously, it has been shown that ablation of GER cells limits the degree of IPhC regeneration, suggesting that the GER harbors precursors that can regenerate lost IPhCs [15]. Using lineage tracing, our study unambiguously shows that GLAST-Cre$^+$ GER cells mitotically regenerate SCs in the organ of Corti after injury. In the previous and current study, proliferation of GER cells was not detected in the damaged *Plp1Cre$^{ERT/+}$*; *Rosa26$^{DTA/+}$* cochleae likely because cell loss was less severe and restricted only to the IPhCs. In contrast, both IPhCs and lateral GER cells were concurrently ablated in the *Lgr5-DTR* model, as *Lgr5* is more broadly expressed. This is evinced by the escalating degrees of damage leading to progressively more proliferation (**S6 Table**). The higher degree of damage in the *Lgr5-DTR* model also likely accounted for the synapse loss in IHCs and profound hearing loss, neither of which was observed using the *Plp1-DTA* model [15].

Despite proliferation and subsequent differentiation into IPhCs, GER cells appeared lineage-restricted and did not give rise to HCs, as noted by the absence of any EdU-labeled HCs. This suggests that GER cells act primarily as progenitors for SCs. In support of this notion, after HC ablation, fate mapping experiments also failed to detect regenerated HCs deriving from the GER [58]. On the other hand, isolated GER cells can proliferate and form HC-like cells [59,60], underscoring their context-dependent ability to act as HC progenitors. In the zebrafish lateral line system, 2 independent studies have demonstrated that SCs in the dorsal/ventral poles proliferate and act as transit amplifying cells and that mantle cells on the periphery divide to replenish SCs after severe injury [61,62]. As mitotic progenitors for SCs, GER cells share features with these SCs and mantle cells from the neuromasts.

In our study, although the IPhCs were replenished, we did not observe a return of Lgr5-EGFP signal. This result contrasts with those obtained with self-renewing organs where Lgr5$^+$ cells reemerge after damage [16,22]. This difference may be because Lgr5 expression in the medial compartment decreases in the neonatal cochlea and regenerated IPhCs have likely matured past this developmental stage [23] and is also in agreement with the diminishing regenerative capacity as the neonatal cochlea matures [12,14]. Furthermore, regenerated IPhCs displayed several molecular markers (e.g., Fabp7, GLAST, Na$^+$/K$^+$ ATPase α-1) and corresponding physiological functions, suggesting some degree of differentiation and maturation. Of note, calcium transients appeared highly active in the damaged cochlea despite a down-regulation of *Ano1*. This is surprising since *Ano1* knockout mice were reported to exhibit significantly fewer spontaneous calcium activities [7]. One possibility is that the hyperactive calcium transients resulted from high levels of extracellular potassium released by degenerating cells. An alternative explanation is that the expanded calcium transients may be as a result of the damage itself. Both of these possibilities warrant further investigation.

## Compartmentalized degeneration/regeneration in the neonatal cochlea

After HC ablation in the neonatal cochlea, mitotic regeneration is limited to the apical turn, and regenerated HCs are short-lived [12,13]. In contrast, mitotic regeneration of IPhCs occurred along the length of the cochlea, and regenerated cells remain present in the mature organ. Interestingly, SCs in the lateral compartment in each cochlear turn also proliferated (**S4 Table**), though to a lesser degree than the medial compartment and regenerated cells underwent delayed degeneration.

Similarly, ablation of Prox1$^+$ SCs in the lateral compartment induced secondary OHC degeneration, without affecting cell survival in the medial compartment [32]. The degeneration of the lateral compartment SCs and OHCs observed in our model may be due to a previously unreported up-regulation of Lgr5 expression in the first and second rows of DCs and outer pillar cells in P2 and P3 Lgr5-DTR mouse cochlea (**S1 Fig**). Alternatively, the degeneration of the third row of DCs may lead to a secondary degeneration of the remaining DCs and outer pillar cells, and subsequently OHCs in a noncell autonomous fashion. Interestingly, the time course of delayed degeneration is similar to that observed following HC ablation, suggesting that the lateral compartment lacks of survival factors following regeneration [12,13]. While the robust proliferation observed in the medial compartment can be attributed to the innate regenerative capacity of GER cells, it is also possible that injury targeting SCs provokes a different and broader response than HC loss alone.

Among the damage-up-regulated genes (clusters 3, 4, 8, and 9), it is notable that *Egr1*-related gene expression overlaps spatially with proliferative cells, particularly in the GER. In the liver, loss of function of *Egr1* partially attenuated regeneration [50]. *Egr4* acts upstream and *Atf3* downstream of *Egr1* in other contexts [63,64]. Whether they play a role in mitotic

regeneration in the damaged cochlea should be of interest in future studies. Of note, cluster 2 consists of genes differentially expressed in input relative to IP samples, with the former collected from whole cochlea and the latter enriched GER cells. As such, cluster 2 genes contained numerous HC genes (**S13 Table**).

During development, both HCs and SCs arise from a common precursor domain [65,66]. Though GER cells express some molecular markers similar to organ of Corti SCs (e.g., Sox2, Jag1), they are morphologically, functionally, and molecularly distinct [18,24]. The current study reveals that GER cells display a unique molecular profile and that they are capable of dividing and migrating laterally to replace lost IPhCs. Regeneration is restricted to the medial compartment with no migration to the lateral compartment detected, implicating that the regenerative and migratory potentials for GER cells may be domain specific.

In summary, our work validates previous studies and further characterizes a hidden progenitor cell population in the GER that may serve as an endogenous source for SC progenitors. The unique translatomes of the quiescent and activated GER cells also set the foundation for future mechanistic studies on mitotic regeneration in the mammalian cochlea.

## Methods

### Mice

The following mouse strains were used: $Lgr5^{DTR-EGFP/+}$ (gift from F. de Sauvage, Genentech) [16], *Atoh1-mCherry* (gift from N. Segil, Univ. Southern Calif.), *Pax2-Cre* (gift from A. Groves, Baylor College of Medicine) [29], $Rosa26^{GCaMP3/+}$ (known as Ai38, Stock #14538, Jackson Laboratory) [67], $GLAST^{CreERT/+}$ (known as Slc1a3-CreERT, Stock #12586, Jackson Laboratory) [28], $Sox2^{CreERT2/+}$ (Stock #17593, Jackson Laboratory) [68], $Rpl22^{HA/+}$ (Stock #11029, Jackson Laboratory) [38], $Plp1^{CreERT/+}$ (Stock #5975, Jackson Laboratory) [25], $Rosa26R^{tdTomato/+}$ (known as Ai14, Stock #7914, Jackson Laboratory) [27], and $Rosa26R^{DTA/+}$ mice [26]. Mice of both genders were used. To ablate cochlear SCs in $Lgr5^{DTR/+}$ mice, diphtheria toxin (0.1 to 4 ng/g, IM or IP, Millipore) was administrated at P1. Saline (0.9% NaCl) treated $Lgr5^{DTR/+}$ mice and diphtheria toxin–treated wild-type mice were used as controls.

To activate Cre recombinase, tamoxifen dissolved in corn oil (0.075 mg/g for $Plp1^{CreERT/+}$ mice and 0.2 mg/g for $GLAST^{CreERT/+}$ mice, IP, Sigma) was administered at P0 to P1. EdU (25 μg/g, IP, Invitrogen) was injected to label proliferative cells. Institutional Animal Care and Use Committee of Stanford University School of Medicine approved all procedures in accordance with NIH guidelines (protocol #18606).

### Genotyping

Genomic DNA was isolated from collected tail tips by adding 180 μl of 50 mM NaOH and incubating at 98°C for 1 hour, followed by the addition of 20 μl of 1 M Tris-HCl. PCR was performed to genotype transgenic mice with the listed primers in **S14 Table**.

### Immunohistochemistry

Methods modified from those previously reported [13]. Briefly, isolated cochleae were fixed in 4% paraformaldehyde (PFA) (in phosphate buffered solution (PBS) (pH 7.4) Electron Microscopy Services) for 40 to 60 minutes at room temperature. Cochleae isolated from P10 or older animals were decalcified in 0.125 M EDTA for 48 hours at 4°C. Cochleae processed for cryosection were immersed overnight in 30% sucrose, then flash frozen in optimal cutting temperature (OCT) compound, and sliced into 10 μm thick sections. Tissues were permeabilized with 0.5% TritonX-100 (in PBS) for 1 hour at room temperature, and then blocked with 5% donkey

serum, 0.1% TritonX-100, 1% bovine serum albumin, and 0.02% sodium azide (NaN3) in PBS (pH 7.4) for 1 hour at room temperature. This was followed by incubation with primary antibodies in the same blocking solution overnight at 4°C. The following day, tissues were washed with PBS 3 times at 5-minute intervals and then incubated with secondary antibodies diluted in PBS containing 0.1% TritonX-100, 1% bovine serum albumin, and 0.02% NaN3 for 2 hours at room temperature. After washing with PBS 3 × 5 minutes, tissues were mounted in ProLong Gold Antifade Mountant (Invitrogen) and coverslipped.

The following primary antibodies were used: rat anti-CD44 (1:200; BD Biosciences), goat anti-CtBP2 (1:500; Santa Cruz Biotechnology), chicken anti-GFP (1:1,000; Aves Labs), rabbit anti-GLAST (1:200; Abcam), mouse anti-GluR2 (1:1,000; Millipore), rabbit anti-Ki67 (1:1,000; Abcam), rat anti-Ki67 (1:400; ThermoFisher Scientific), rabbit anti-Myosin7a (1:1,000; Proteus Bioscience), mouse anti-Myosin7a (1:1,000; Developmental Studies Hybridoma Bank), mouse anti-Na$^+$/K$^+$ ATPase α-1, α6F (1:500; Developmental Studies Hybridoma Bank), goat anti-Sox2 (1:200; Santa Cruz Biotechnology or R&D Systems), goat anti-Sparcl1 (1:500; R&D Systems), mouse anti-Tuj1 (1:1,000; Neuromics), rabbit anti-Fabp7 (1:200; Abcam), and rabbit anti-Vglut3 (1:1,000; Synaptic Systems). Alexa Fluor secondary antibodies (405, 488, 546, or 647, 1:500; Invitrogen or Jackson ImmunoResearch) and Rhodamine Red-X (Jackson ImmunoResearch) were used. Fluorescent-conjugated Phalloidin (1:1,000; Invitrogen), DAPI (1:10,000; Invitrogen), and Click-iT EdU Alexa Fluor 555 or 647 Imaging Kit per manufacturer's instructions (Invitrogen) were also used. Key resources were listed in **S15 Table**.

## Imaging and cell quantification

Whole mount cochleae were captured as z-stack images using LSM700 confocal microscope (Zeiss). Image analyses were performed using Zen Software (Zeiss), ImageJ (NIH), and Photoshop CS6 (Adobe Systems). Cells were quantified from z-stack confocal images (1 μm intervals) of the sensory epithelia from P3 to P21 mice using ImageJ (NIH) and Zen lite (Zeiss) software using previously described methods [15]. For cochleae aged P3 to P10, IPhCs were identified as 2 rows of Sox2$^+$ cells immediately medial to IHCs with nuclei located below those of IHCs. Sox2$^+$ cells located at or higher than that of IHCs were considered GER cells and excluded. For cochleae aged P14 to P21, IPhCs were identified as Sox2$^+$ or GLAST$^+$ cells immediately medial to IHCs. Pillar cells and DCs were counted together as Sox2$^+$ cells lateral to IHCs with nuclei located below those of OHCs. Representative confocal images were taken from the apical, middle, and basal turns of each cochlea.

## Ribosome immunoprecipitation and RNA extraction

Cochleae were isolated from P4 *Lgr5$^{DTR-EGFP/+}$; GLAST$^{CreERT/+}$; Rpl$^{22HA/+}$* and *GLAST$^{CreERT/+}$; Rpl$^{22HA/+}$* mice (spiral ganglia, lateral wall, and Reissner's membrane were removed) in cold Hanks' Balanced Salt Solution (HBSS) and then immediately flash-frozen using liquid nitrogen prior to RNA isolation. About 2 to 8 cochleae were pooled and processed for ribosome IP followed by RNA extraction as described [40]. Briefly, cochlear ducts were homogenized and incubated with 5 μg purified anti-HA.11 (BioLegend) for 6 hours at 4°C before the addition of the equivalent of 300 μl Dynabeads protein G (ThermoFisher Scientific) and further incubation overnight. Additionally, 5% of the homogenate was kept before addition of the antibody to be used as input. RNA was extracted from the IP and input samples using the RNeasy Plus Micro kit (Qiagen) following the manufacturer's instructions. RNA quality was confirmed with a BioAnalyzer 2100 picochip (Agilent Technologies) performed at the Genomics Core Facility in the Center for Innovative Biomedical Resources (University of Maryland School of Medicine). All RNA integrity numbers (RINs) were above 8.

## Assessment of IP efficiency by real-time RT-PCR

Cell type–specific RNA enrichment following the IP was assessed by reverse transcription (Maxima First Strand cDNA Synthesis Kit for RT-qPCR, ThermoFisher Scientific) using equivalent amounts of IP and corresponding input samples, followed by real-time PCR (Maxima SYBR Green/ROX qPCR Master Mix, ThermoFisher Scientific) on a StepOnePlus Real-Time PCR System (Applied Biosystems). The primers used are listed in **S14 Table**.

## RNAseq of RiboTag samples

Four individual biological replicates were used for sequencing, with each replicate comprised of 8 cochleae from 4 mice. About 1 to 3 ng of RNA from the input and IP samples were used as template for library preparation with the SMART-Seq v4 Ultra Low Input RNA Kit for Sequencing. Libraries were sequenced on an Illumina HiSeq 4000 at 75 paired-end read length and a depth of sequencing of 74 to 100 million reads per sample. Library preparation and sequencing were performed at the Institute for Genome Sciences, University of Maryland School of Medicine. The RNA sequencing data generated in this paper are available from GEO with accession number GEO: GSE135728. The data are also available at gEAR, a gene Expression Analysis Resource (https://umgear.org), via PERMA-LINK https://umgear.org/p?s= 3296013a.

## RNAseq normalization and expression analysis

Quality of Fastq files were evaluated by using FastQC. Reads were aligned to the mouse genome (Mus_musculus.GRCm38) using HiSat (version HISAT2-2.0.4) [69], and the number of reads that aligned to the coding regions was determined using HTSeq [70]. Approximately 171,121 genes were assessed for significant differential expression using DESeq2 with an FDR value $\leq$0.05 [71]. The RiboTag immunoprecipitated samples and their corresponding input were compared to generate an EF. Transcripts were considered enriched in the IP if the log2 of the ratio between the IP and the input was $\geq$2. For DEGs between the IP samples, the criteria of LFC $\geq$1 between DTR IP and Control IP samples was used. Additionally, to avoid overinflation of fold change, all the values were quantile normalized CPM, and values lower than 10% of the value of the dataset were replaced with the 10th quantile. Furthermore, only genes with cutoff >1.5 (min(group1) / max(group2)) were considered for differential expression or enrichment. For hierarchical clustering, similarity between samples was measured using Pearson correlation, as such the samples' distance was calculated as 1—Pearson correlation coefficient for each sample pair. The heat map was generated using the R packages pheatmap and RColorBrewer. PCA was done using DESeq2. Gene Ontology was performed using DAVID [72,73].

## NanoString reactions and analysis

The nCounter technology from NanoString measure gene expression at the RNA level. NanoString reactions were performed in technical duplicates on the 4 biological replicates used for RNA-seq plus 1 independent fifth replicate. About 1 ng of RNA from input and IP samples was used for preamplification with the nCounter Low RNA Input Kit (NanoString Technologies, WA) to obtain sufficient cDNA to be run on the Counter platform (NanoString Technologies) (Primers for preamplification are listed in **S14 Table**). A Custom CodeSets for 28 targets including 4 housekeeping genes was designed (probes listed in **S14 Table**), and the samples were run with the nCounter Master Kit (NanoString Technologies) following the manufacturer's protocol. Preamplification, nanoString reactions, and quality check steps were performed at the Institute for Genome Sciences, University of Maryland School of Medicine.

Normalization to housekeeping genes and data analysis was performed using the nSolver 4.0 analysis software (NanoString).

## Clustering analysis

Prior to clustering, conditional quantile normalization (CQN) was performed to correct for sample-specific gene length effect in the RiboTag IP samples [74]. Genes were included in the analysis only if (1) their expression level was readily detected in at least one of the biological conditions (i.e., their expression was at least 0.5 cpm in all replicate samples of at least one of the 4 conditions); (2) showed significant differential expression (FDR < 0.05 and FC > 1.5) either between "cell type" (i.e., between input and IP samples) or in response to DT-induced damage; and (3) expression levels of a DEG were fully separated between the 2 conditions the DE was called. Overall, 5,095 genes met these criteria and were subjected to cluster analysis to detect the main expression patterns in the dataset. Cluster analysis was done using the CLICK algorithm implemented in the EXPANDER package [43].

## Live cell imaging

Calcium imaging using *Pax2-Cre*; *R26*$^{GCaMP3/+}$ mice was performed using methods modified from a previous report [7]. Cochleae were harvested from P7 mice carrying the following alleles: (1) *Atoh1-mCherry*; *Pax2-Cre*; *R26*$^{GCaMP3/+}$; (2) *Lgr5*$^{DTR-EGFP/+}$; *Atoh1-mCherry*; *Pax2-Cre*; *R26*$^{GCaMP3/+}$; (3) *Pax2-Cre*; *R26*$^{GCaMP3/+}$; and (4) *Lgr5*$^{DTR-EGFP/+}$; *Pax2-Cre*; *R26*$^{GCaMP3/+}$. The apical turn was isolated with modiolus and stria vascularis removed and imaged in glass chamber containing 100 μl HBSS with calcium chloride (1.26 mM) and magnesium chloride (0.49 mM) (ThermoFisher Scientific). Images were captured at the focal point of HC nuclei at a frequency of once per second for 3 to 5 minutes using confocal microscopy (LSM 700; Zeiss). A 20X objective (NA0.8), 488 nm and 555 nm laser illumination, and LP 560 and SP 555 filters were used. Laser power and gain were consistent across samples. For quantifying Ca$^{2+}$ transients in IPhCs, only events that spanned more than 3 IHCs were analyzed. The frequency, area, and intensity of spontaneous Ca$^{2+}$ transients (GFP signals) in SCs were measured using ImageJ software (NIH). The average intensity 2 seconds before the peak intensity of an event was considered baseline fluorescence (F), and the intensity value of the event was calculated as the difference between the peak fluorescence and baseline / baseline (ΔF/F).

## In situ hybridization

Harvested tissues were fixed in 4% PFA overnight at 4°C, embedded for cryosections, and prepared as 10 μm sections. Tissue sections were hybridized with probes from Advanced Cell Diagnostics (ACDbio) and counterstained with hematoxylin (Sigma-Aldrich) according to the manufacturer's instructions for fixed frozen sections with colorimetric detection. Briefly, sections were washed once in PBS for 5 minutes and then treated with H$_2$O$_2$ for 10 minutes. Next, sections were permeabilized using target retrieval reagent (ACDbio) and proteinase before hybridization. Probes used were as follows: *DapB* (Cat: 310043), *Polr2a* (Cat: 312471), *Igf2* (Cat: 437671), *Igf2bp1* (Cat: 451921), *Egr1* (Cat: 423371), *Egr4* (Cat: 553851), and *Atf3* (Cat: 426891) (Advanced Cell Diagnostics). Undamaged and damaged cochleae were processed in parallel, with sections collected on the same slide and subjected to mRNA detection under identical conditions.

## Auditory physiology measurements

ABRs and DPOAEs were measured as previously described [75]. Briefly, P21 mice were anesthetized with a ketamine/xylazine mixture (100 mg/kg ketamine and 10 mg/kg xylazine, IP)

and placed on a heating pad at 37˚C. ABRs were recorded from a needle electrode located inferior to the tympanic bulla, referenced to an electrode on the vertex of the head, and a ground electrode was placed at the hind limb. Tone pip stimuli were delivered with frequencies ranging from 4 to 46 kHz (4.0, 5.7, 8.0, 11.3, 16.0, 23.0, 31.9, 46.1 kHz) up to 80 dB sound pressure level (SPL) in 10 dB steps. At each frequency and SPL, 260 trials were tested and averaged. DPOAEs were recorded with a probe tip microphone placed in the auditory canal. Two 1-second sine wave tones of different frequencies ($F2 = 1.22 \times F1$) were used as the sound stimuli. F2 ranged from 4 to 46 kHz (4.0, 5.7, 8.0, 11.3, 16.0, 23.0, 31.9, 46.1 kHz), and the 2 tones were from 20 to 80 dB SPL in 10 dB steps. The amplitude of the cubic distortion product was recorded at $2 \times F1-F2$. The threshold was calculated as a cross point of DPOAE signal with the noise floor level above 3 standard deviations at each frequency. For statistical analyses of both ABR and DPOAE thresholds, a lack of a response was defined as the highest sound level, 80 dB SPL.

## Statistical analyses

Statistical analyses were performed using Microsoft Excel (Microsoft) and GraphPad Prism 7.03 (GraphPad). For comparison of 2 groups, Student $t$ test and Mann–Whitney test were used. One-way ANOVA was used when comparing more than 2 groups, and a two-way ANOVA was used for comparison with 2 independent variables. $p < 0.05$ was considered statistically significant.

## Supporting information

**S1 Fig. Spatiotemporal expression of Lgr5-EGFP during cochlear degeneration in Lgr5-DTR mice.** (A) Cartoon depiction of Lgr5-EGFP expression in the P1 *Lgr5*$^{DTR/+}$ cochlea. (B, F) Undamaged, saline-treated *Lgr5*$^{DTR/+}$ (control) cochleae showed Lgr5-EGFP expression in IPhCs, inner PCs, and the third row of DCs in the apical turn at P2 and P3. (C-E, G-I) In the DT-treated *Lgr5*$^{DTR/+}$ cochleae, the first and second rows of DCs and outer PCs ectopically expressed Lgr5-EGFP with partial loss in all 3 turns at both ages. DC, Deiters' cell; DT, diphtheria toxin; GER, greater epithelial ridge; IHC, inner hair cell; IPhC, inner phalangeal cell; LER, lesser epithelial ridge; OHC, outer hair cell; PC, pillar cell.
(TIF)

**S2 Fig. Degeneration and regeneration of cochlear cells in *Lgr5-DTR* mice.** (A, C, E, G, I, K) Shown are representative confocal images from middle and basal turn of the saline-treated *Lgr5*$^{DTR/+}$ (undamaged) cochleae. Lgr5-EGFP signals were detected in the lateral GER, IPhCs, inner PCs, and the third row of DCs at P4 and P7, while at P21, expression was restricted to the third row of DCs at P21. No pyknotic nuclei were detected at P4. (B, D) In the P4 DT-treated *Lgr5*$^{DTR/+}$ cochlea, most Sox2$^+$ cells were lost in the IPhC and PC/DC regions. (B", D") Many pyknotic nuclei were observed in the IPhC region (arrowheads). (F, H, J, L) In each turn of the P7 and P21 DT-treated *Lgr5*$^{DTR/+}$ cochlea, IPhCs were present and at cell densities comparable to controls. However, Sox2$^+$ SCs in the PC/DC region remained depleted. (M, N) Quantification of Sox2$^+$ or Myosin7a$^+$ cells (per 160 µm) in the apical, middle, or basal cochleae (normalized to control). Normalized Sox2$^+$ PC/DC counts were reduced by P4 and partially regenerated at P7, followed by a delayed and progressive degeneration at P14 and P21 in the apical and middle turns ($n = 6$ at P4, $n = 8$ at P7, $n = 6$ at P14, and $n = 8$ at P21). In the basal turn, normalized Sox2$^+$ PC/DC counts gradually decreased. There were no detectable changes in normalized Myosin7a$^+$ IHC counts. Dashed lines highlight IPhC region. $n = 4$–8. See S1 Data for M and N. DC, Deiters' cell; DT, diphtheria toxin; GER, greater epithelial ridge;

IHC, inner hair cell; IPhC, inner phalangeal cell; PC, pillar cell; SC, supporting cell.
(TIF)

**S3 Fig. Mitotic regeneration of IPhCs.** (A-L) Representative images of the apical, middle, and basal turns of DT-treated wild-type (control) undamaged cochleae showing no EdU⁺ or Ki67⁺ Sox2⁺ SCs at P14 or P21. The IPhC region is outlined by dashed lines. In the DT-treated P14 and P21 *Lgr5*$^{DTR/+}$ cochlea, many EdU⁺ Sox2⁺ cells remained in IPhC region in all 3 turns. No Ki67⁺ Sox2⁺ cells were detected in the IPhC or PC/DC regions. (M-P) Quantification of EdU⁺ Sox2⁺ cells in the apical, middle, and basal turns of control (DT-treated wild type) and damaged (DT-treated *Lgr5*$^{DTR/+}$) cochleae at P14 and P21. Wild-type cochleae had almost no EdU⁺ Sox2⁺ cells at both ages examined. Conversely, there were significantly more EdU⁺ Sox2⁺ IPhCs in the damaged cochleae with an apex-to-base gradient at both P14 and P21. Some EdU⁺ Sox2⁺ cells survived in PC-DC regions in the damaged cochleae in all 3 turns at both ages. (Q) Schematic showing DT administration to P1 wild-type or *Lgr5*$^{DTR/+}$ mice. EdU was injected daily from P7 to P9, and cochleae were examined at P10. (R, S) In both DT-treated wild-type (control) and DT-treated *Lgr5*$^{DTR/+}$ cochleae, there were no EdU⁺ or Ki67⁺ Sox2⁺ cells. Data represent mean ± SD. $^{*}p < 0.05$, $^{***}p < 0.001$ (two-way ANOVA with Tukey's multiple comparisons test). $n = 3$–5. See S1 Data for M-P. DC, Deiters' cell; DT, diphtheria toxin; GER, greater epithelial ridge; IPhC, inner phalangeal cell; PC, pillar cell; SC, supporting cell.
(TIF)

**S4 Fig. Minimal proliferation after ablation of IPhCs in *Plp1-DTA* mice.** (A) Tamoxifen was injected into P0 and P1 *Plp1*$^{CreERT/+}$; *R26R*$^{tdTomato/+}$; *R26R*$^{DTA/+}$ mice. *Plp1*$^{CreERT/+}$; *R26R*$^{tdTomato/+}$ mice served as undamaged controls. EdU was injected daily from P3 to P4, and cochleae were examined at P4. (B-D) Representative images of the each turn of control cochleae showing Plp1-tdTomato⁺ Sox2⁺ IPhCs (dashed lines). No EdU-labeled Sox2⁺ cells were detected. (E-G) In damaged cochleae, fewer Plp1-tdTomato⁺ Sox2⁺ IPhCs were detected, and rare EdU⁺ Sox2⁺ cells were detected in the GER (arrowheads) in the apical and middle turns. (H) Quantification showing a significant reduction of Plp1-tdTomato⁺ in the middle and basal turns. (I) EdU⁺ Sox2⁺ cells in the GER were rarely found, and the number was not significantly different from control cochleae. Data represent mean ± SD. $^{***}p < 0.001$ (two-way ANOVA with Tukey's multiple comparisons test). $n = 3$–5. See S1 Data for H and I. DC, Deiters' cell; GER, greater epithelial ridge; IHC, inner hair cell; IPhC, inner phalangeal cell; OHC, outer hair cell; PC, pillar cell.
(TIF)

**S5 Fig. IPhCs do not self-regenerate.** (A) Plp1-tdTomato expression in the P3 *Plp1*$^{CreERT/+}$; *R26R*$^{tdTomato/+}$ cochlea. (B) Schematic of the experimental paradigm: DT and tamoxifen were injected into the P1 *Plp1*$^{CreERT/+}$; *R26R*$^{tdTomato/+}$ (control) or *Lgr5*$^{DTR/+}$; *Sox2*$^{CreERT2/+}$; *R26R*$^{tdTomato/+}$ (damage) mice. EdU was injected daily from P3 to P5, and cochleae were examined at P3, P7, or P14. (C-E) Representative images of the apical turn of control cochleae showing Plp1-tdTomato⁺ Sox2⁺ or SCs at P3 and P7. At P14, Plp1-tdTomato⁺ IPhCs expressed GLAST. IPhC region outlined by dashed lines. (F-H) In damaged cochleae, Plp1-tdTomato⁺ Sox2⁺ cells were not detected in the IPhC region or in the GER at any age. Conversely, EdU⁺ Sox2⁺ Plp1-tdTomato-negative cells were detected in the GER at P3. At P7 and P14, EdU⁺ Sox2⁺ Plp1-tdTomato⁺ cells were not found in the GER/IS or IPhC regions. Many EdU⁺ Sox2⁺ Plp1-tdTomato-negative cells were detected in the IPhC region at P7 and P14. Orthogonal views shown in C'-H'. (I) Quantification of Plp1-tdTomato⁺ Sox2⁺ SCs in the apical turn. (J, K) Quantification of EdU⁺ Sox2⁺ SCs in the apical turn. In damaged cochleae, there were no Plp1-tdTomato⁺ Sox2⁺ SCs in the GER or in the IPhC region at any age. There was, however,

an increase in Sox2$^+$ EdU$^+$ Plp1-tdTomato-negative cells in the GER peaking at P7, followed by a reduction at P14. There were no Sox2$^+$ EdU$^+$ Plp1-tdTomato-negative cells in the IPhC region at P3, but many at P7 and P14. Data represent mean ± S.D. $^{***}p < 0.001$ (two-way ANOVA with Tukey's multiple comparisons test). $n = 4$. See S1 Data for I-K. DC, Deiters' cell; DT, diphtheria toxin; GER, greater epithelial ridge; IHC, inner hair cell; IPhC, inner phalangeal cell; IS, inner sulcus; LER, lesser epithelial ridge; OHC, outer hair cell; PC, pillar cell; SC, supporting cell.
(TIF)

**S6 Fig. Molecular and physiological properties of regenerated juvenile and mature cochlea.** (A) In undamaged P7 (control) cochleae, CD44 is expressed in outer PCs, Claudius cells, and the LER. Representative images of the apical turn are shown. IPhC region is labeled by dashed lines. (B) In the P7 DT-treated $Lgr5^{DTR/+}$ cochlea, Sox2$^+$ SCs in the PC/DC region have degenerated, with CD44$^+$ Claudius cells and LER appearing grossly intact. (C, D) In both the undamaged and damaged P14 cochlea, Vglut3$^+$ IHCs were surrounded by Na$^+$/K$^+$ ATPase α-1-expressing, Sox2$^+$ IPhCs. (E) GLAST expression (membranous) of Sox2$^+$ IPhCs in the P14 undamaged cochleae. IPhC region is outlined by dashed lines. (F) In the P14 damaged cochlea, all IPhCs expressed GLAST, and most were EdU-labeled. (G) In the P14 control cochlea, there were no EdU$^+$ GLAST-tdTomato$^+$ SCs in the inner sulcus. GLAST$^+$ IPhCs were mostly GLAST-tdTomato$^+$ but not EdU$^+$. (H) In the P14 damage cochlea, many EdU$^+$ GLAST-tdTomato$^+$ GLAST$^+$ IPhCs were detected. Orthogonal view shown in H'. (I, J) P21 DT-treated $Lgr5^{DTR/+}$ mice had higher ABRs and DPOAEs thresholds at all frequencies than controls. Data represent mean ± S.D. $^{***}p < 0.001$ (two-way ANOVA with Tukey's multiple comparisons test). $n = 7$–8. See S1 Data for I and J. ABR, auditory brainstem response; DC, Deiters' cell; DPOAE, distortion product otoacoustic emission; DT, diphtheria toxin; GER, greater epithelial ridge; IHC, inner hair cell; IPhC, inner phalangeal cell; LER, lesser epithelial ridge; OHC, outer hair cell; PC, pillar cell; SC, supporting cell; SPL, sound pressure level.
(TIF)

**S7 Fig. Gene expression of GLAST-Cre$^+$ cells.** (A) Validation of a selection of genes (*Dlx5*, *Igf2*, *Socs1*, and *Socs3*) enriched in GLAST-Cre$^+$ samples using nCounter. (B, C) *In situ* hybridization showing *Igf2bp1* and *Igf2* expression in the GER of control and damage P4 cochlea from the apical and middle turns. (D) Positive (*Polra*) and negative (*DapB*) controls shown. GER (blue bracket); OC (black bracket). Data represent mean ± SD. $^*p < 0.05$ (Student $t$ test). See S1 Data for A. DTR, diphtheria toxin receptor; GER, greater epithelial ridge; IP, immunoprecipitation; OR, organ of Corti.
(TIF)

**S8 Fig. Differential gene expression in the GER after ablation of Lgr5$^+$ cells.** (A) qPCR showing *MKi67* mRNA expression from individual control and DTR samples. (B-D) nCounter was used to valid another 12 DEGs as a result of damage. Eight were successfully validated (*Lfng*, *Ppp2r2b*, *Trh*, *Cenpf*, *Ccnb2*, *Cdk1*, *Cacna1c*, and *Iqck*), and 4 (*Bmp4*, *Fgfrl1*, *Junb*, and *Id4*) were tested and not validated. (E-G) *Egr1*, *Egr4*, *Atf3* mRNA expression was minimally expressed in the undamaged cochleae. Expression was robust in the lateral GER and OC after damage. Shown are sections from the apical and middle turns. GER (blue bracket); OC (black bracket); data represent mean ± SD. $^*p < 0.05$, $^{**}p < 0.01$, $^{***}p < 0.001$ (Student $t$ test). See S1 Data for A-D. DEG, differentially expressed gene; DTR, diphtheria toxin receptor; GER, greater epithelial ridge; IP, immunoprecipitation; OR, organ of Corti.
(TIF)

**S9 Fig. Schematic of mitotic regeneration by GER cells. GER, greater epithelial ridge.**
(TIF)

**S1 Movie. Time-lapse imaging of spontaneous activity in control and damaged cochlea.**
Live imaging of P7 cochleae from DT-injected *Lgr5*$^{DTR/+}$; *Atoh1-mCherry*; *Pax2-Cre*;
*R26*$^{GCaMP3/+}$ and *Atoh1-mCherry*; *Pax2-Cre*; *R26*$^{GCaMP3/+}$ mice. Apical turn shown. DT, diph-
theria toxin.
(MOV)

**S1 Table. Quantification of Sox2$^+$ SCs and Myosin7a$^+$ HCs. HC, hair cell; SC, supporting
cell.**
(XLSX)

**S2 Table. Pyknotic nuclei counts in control and damaged cochleae.**
(XLSX)

**S3 Table. Quantification of Ki67$^+$ Sox2$^+$ SCs. SC, supporting cell.**
(XLSX)

**S4 Table. Quantification of EdU$^+$ Sox2$^+$ SCs. SC, supporting cell.**
(XLSX)

**S5 Table. Dose response relationship between DT and Sox2$^+$ cell counts. DT, diphtheria
toxin.**
(XLSX)

**S6 Table. Dose response relationship between DT and proliferation. DT, diphtheria toxin.**
(XLSX)

**S7 Table. Quantification of fate-mapped and total IPhCs. IPhC, inner phalangeal cell.**
(XLSX)

**S8 Table. Quantification of EdU$^+$ GLAST-tdTomato$^+$ IPhC and GER cells. GER, greater
epithelial ridge; IPhC, inner phalangeal cell.**
(XLSX)

**S9 Table. Analyses of P14 and P21 normal and damaged cochleae.**
(XLSX)

**S10 Table. Genes that contribute to PCA and GO terms of enriched and differentially
expressed genes. GO, gene ontology; PCA, principal component analysis.**
(XLSX)

**S11 Table. Highly enriched genes in Glast-Cre$^+$ cells.**
(XLSX)

**S12 Table. DEGs between control and damaged Glast-Cre$^+$ cells. DEG, differentially
expressed gene.**
(XLSX)

**S13 Table. Genes associated with cluster analyses.**
(XLSX)

**S14 Table. Primer sequences.**
(XLSX)

**S15 Table. Key resources.**
(XLSX)

**S1 Data. Raw data for main and supplementary figures.**
(XLSX)

## Acknowledgments

We thank T. Nicolson, S. Heller, A. Groves, J. Zuo, and our laboratory for insightful comments on the manuscript; R. Nusse and A. Ricci for fruitful discussion; F. Sauvage, A. Groves, and N. Segil for mouse sharing; C. Gralapp for illustration; and P. Minhas, R. Yamashita, S. Billings, and W. Dong for excellent technical support.

## Author Contributions

**Conceptualization:** Tomokatsu Udagawa, Julia M. Abitbol, Ronna Hertzano, Alan G. Cheng.

**Data curation:** Tomokatsu Udagawa, Julia M. Abitbol, Alan G. Cheng.

**Formal analysis:** Tomokatsu Udagawa, Patrick J. Atkinson, Beatrice Milon, Julia M. Abitbol, Yang Song, Michal Sperber, Mirko Scheibinger, Ronna Hertzano, Alan G. Cheng.

**Funding acquisition:** Tomokatsu Udagawa, Patrick J. Atkinson, Ronna Hertzano, Alan G. Cheng.

**Investigation:** Patrick J. Atkinson, Beatrice Milon, Julia M. Abitbol, Michal Sperber, Elvis Huarcaya Najarro, Ronna Hertzano, Alan G. Cheng.

**Methodology:** Mirko Scheibinger.

**Project administration:** Patrick J. Atkinson, Alan G. Cheng.

**Validation:** Patrick J. Atkinson, Elvis Huarcaya Najarro.

**Writing – original draft:** Tomokatsu Udagawa, Patrick J. Atkinson, Alan G. Cheng.

**Writing – review & editing:** Tomokatsu Udagawa, Patrick J. Atkinson, Beatrice Milon, Julia M. Abitbol, Yang Song, Michal Sperber, Elvis Huarcaya Najarro, Mirko Scheibinger, Ran Elkon, Ronna Hertzano, Alan G. Cheng.

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
