## [Editor Report · Decision Letter 0]

9 Apr 2021

Dear Dr Cheng, 

Thank you for submitting your manuscript entitled "Translatomic analysis of mitotic cochlear progenitors after ablation of Lgr5+ cells" for consideration as a Research Article by PLOS Biology.

Your manuscript has now been evaluated by the PLOS Biology editorial staff as well as by an academic editor with relevant expertise and I am writing to let you know that we would like to send your submission out for external peer review.

Please re-submit your manuscript within two working days, i.e. by Apr 13 2021 11:59PM.

Kind regards,

Lucas Smith, Ph.D.,

Associate Editor

PLOS Biology

---

## [Decision Letter · Decision Letter 1]

19 May 2021

Dear Dr Cheng,

Thank you very much for submitting your manuscript "Translatomic analysis of mitotic cochlear progenitors after ablation of Lgr5+ cells" for consideration as a Research Article at PLOS Biology. Your manuscript has been evaluated by the PLOS Biology editors, an Academic Editor with relevant expertise, and by several independent reviewers.

The reviews of your manuscript are appended below. While the reviewers think that the study is interesting and well done, they also raise a number of important concerns which will need to be addressed before we can consider your manuscript for publication at PLOS Biology. The reviewers question aspects of novelty, they require new and more convincing analyses to support the conclusions, and suggest a general reorganization of the paper.

If you feel you can make the changes required to thoroughly address the reviewer concerns, with new data where appropriate, we are willing to consider a revised version of the paper. We appreciate that this would be a large undertaking, and we would would also understand if you prefer to send the paper elsewhere. We cannot make any decision about publication until we have seen the revised manuscript and your response to the reviewers' comments. Your revised manuscript is also likely to be sent for further evaluation by the reviewers.

We expect to receive your revised manuscript within 3 months. 

**IMPORTANT - SUBMITTING YOUR REVISION**

*Re-submission Checklist*

*Published Peer Review*

*PLOS Data Policy*

*Blot and Gel Data Policy*

Sincerely,

Lucas Smith

Associate Editor

PLOS Biology

lsmith@plos.org

REVIEWS:

Reviewer #1: The authors perform a study to selectively ablate Lgr5+ cells in the neonatal cochlea. Lgr5+ in the GER was initially thought to be responsible for maintaining stemness in the post-mitotic cochlea; however, the authors now describe a different population of cells in the GER contribute to regeneration. The idea that the GER holds some potential to proliferate has been known for a while and is not novel though. The mechanisms that regulate this stemness has yet to be unveiled. The authors use Lgr5-DTR to study spontaneous regeneration (has been demonstrated before in several studies) of IPhCs from the Lgr5- cell population in the GER. The study is well-done but there is quite a bit of distracting data/ claims that are unresolved, difficult to consolidate and take away from the main crux of the story. As a result, the manuscript, as written, feels descriptive without functional testing of these ideas. 

Major points:

1. Loss of OHCs, but only 3rd DC has Lgr5. Is Lgr5-DTR too toxic that it's killing neighboring cells? Some explanation of why other cells in the lateral compartment are affected, is warranted. 

2. Figure 1- IHCs look impacted both in number and morphology. Do any of the HCs arise from the proliferated cells from the GER? i.e,. EdU positive?

3. Figures 1E"-I". What am I supposed to take away from these images? Not described in the text. 

4. Figure 1L there are fewer EdU positive cells at P7, P14 and P21 than at P4, suggesting that they died off. But the authors say that they survive. Image does not support this statement. 

5. Figure 2A- Very difficult to make out Edu and Ki67. Use different color scheme. Preferably, one that's colorblind friendly. 

6. Figure 3- Pax2Cre; Lgr5-DTR, Atoh1-mCherry, Rosa26R-GCaMP3. If Lgr5 cells die off as early as E8.5 with Pax2-Cre, how is there a cochlea at all?

7. Figure 5- What is E'-J' supposed to tell me? No explanation.

8. Page 12, line 7: 'Collectively, these results suggest that regenerated IPhCs are partially mature/functional and remained viable in the mature organ.' I am not sure if the authors can make this conclusion. 

9. Authors present a number of associated changes that I am not sure how they fit in:

a. Ca2+ transients- does this relate to regeneration of IPhCs? Don't large Ca2+ waves suggest damage as opposed to any regenerative response? No functional data to test a proliferative response. Loss of Pannexin and Connexin is likely a consequence of losing cells. The authors suggest that they partake in proliferation in the GER, but there's no data to support this. Other than showing that there are associated changes, their role in regeneration is not well supported. 

b. Neurite innervation and reduced CtBP2 synapses- I only see damage. Not sure what I am supposed to make of this data and how it related to the main story of IPhC regeneration from GER. It doesn't add very much to the paper or its hypotheses.

c. The authors find a number of genes that were differentially regulated post DTR but does not mean they are associated with the limited regeneration of IPhCs. None were tested. How relevant they are, are yet to be determined at this stage. Thus, making the paper descriptive in nature.

Minor points:

11. Figure 1A- Green Lgr5 is difficult to see. Show monochromatic image for Lgr5.

12. Page 13, line 20: P3 Sox2-tdTomato-DTR cochleae. 

Is this Lgr5-DTR?

13. Page 14, line 5: "…..Sox2-expressing cells in the GER serve as mitotic progenitors capable of replenishing lost IPhCs." Only up to a certain point though. Regenerative potential is lost after apoptosis in the GER.

Reviewer #2: The manuscript by Udagawa et al., describes the regeneration of one subtype of SCs (inner phalangeal cells) after several SC subtypes were killed using the Lgr5-DTR mouse model and injection of DT at P1. Killing of Lgr5+ SCs triggered a mitotic response in the GER that was dependent on the number of SCs killed. Fate-mapping and EdU experiments demonstrated that regenerated inner phalangeal cells were derived from the GER cells. Regenerated cells expressed markers of inner phalangeal cells, appeared functional since they displayed calcium transients, and survived into adulthood. Using the RiboTag system with a CreER specific to the GER cells, the authors also report gene clusters that were altered in different ways after damage.

Overall the manuscript is well written and the experiments are sound with appropriate controls. While this work greatly expands upon what was known previously, I don't think the paper gives enough credit to findings made by Mellado Lagarde et al., 2014 which also concluded that the GER cells were the source of the regenerated inner phalangeal cells. There are also several points that need clarification. See below for major and minor comments.

Major comments

1) Previous work by Mellado Lagarde et al., 2014 is cited however I don't think that enough credit is given to this paper. Mellado Lagarde et al., used 3 different CreER lines to express DTA, killing inner phalangeal cells, border cells, and in some cases also GER cells. They demonstrated that inner phalangeal cells were regenerated from neighboring GER cells (while fate-mapping was not done, the conclusion was still supported by other experiments). In addition, one of the models they used was Lgr5-CreER, which should give the same result as the Lgr5-DTR model used here. There are still many unique findings in this new manuscript (the dose response of SC death, calcium transient recordings, the fate-mapping, and of course the gene expression analysis) that make this work significant and novel. It's just not the 1st to identify GER cells as progenitors, thus, I suggest changing how the study is introduced in the Abstract and Introduction and rewriting the first and last paragraph of the Discussion to focus on the novel findings in this study.

2) I don't find the Connexin26 staining in Figure 4K very convincing. The staining in the inner phalangeal region looks like background especially when compared to the Connexin26 staining at P7 in Figure 3C-D. I also don't understand what is meant by the "plaque-like expression pattern" (page 18)?

3) I don't understand the point of the Sox2-CreER fate-mapping experiments. It does not add anything new to the paper that isn't gleaned from fate-mapping with the GLAST-CreER line. Figure 2 showed that the EdU+ cells are in the GER before the inner phalangeal cells are regenerated, so that's the lead in to using GLAST-CreER for the fate-mapping. Also there is the confounding factor of Sox2 haploinsufficiency in the Sox2-CreER line that could effect the proliferation seen after Lgr5+ cells are ablated.

Minor comments

1) Abstract: Why is the Egr1 gene highlighted in the Abstract and Introduction? There were hundreds of genes changed after ablation of Lgr5+ cells so it sees odd to mention just 1 gene in these 2 locations. What makes it special?

2) Results bottom of page 11: When introducing the Lgr5-DTR mouse, please state that it is a knockin where the endogenous Lgr5 regulatory elements control expression of eGFP and DTR. Similarly when mentioning the Lgr5-CreER line a few sentences later, please state that it is a knockin line and eGFP is the reporter that you refer to (not just "reporter").

3) Figure 1A would be improved if the GFP channel was also shown alone.

4) Figure 1G', I' and K' shows that there is outer HC loss too but this is not mentioned in the Results section until much later in the paper. If you are going to show these images in Figure 1, the HC loss needs to be mentioned here too.

5) Results top of page 13: Figure 1L shows that inner phalangeal cell counts are similar in control and DTR samples at P7, P14 and P21 but P7 was omitted here.

6) Results middle of page 13: please clarify that EdU labels cells that are actively dividing when the EdU injections are given, whereas the Ki67 labels cells outside of G0 at the time of sample fixation.

7) Results top of page 16: The authors conclude that the regenerated cells are inner phalangeal cells based on expression of pannexin1, jagged1 and connexin26. However these 3 genes are also expressed in the GER so I don't think this conclusion is accurate.

8) Figure 4E and 4G would be improved if the myo7a/Vglut3 channel was not present or additional panels without these were added. With the max projections presented as is, it looks like the GLAST/NA-K ATPase staining is surrounding the inner HCs. 

9) Figure 4P-Q: Need to add a graph of the synapse counts to show the significant differences. 

10) Results top of page 19: In addition to what is written as a conclusion of this paragraph, the data also shows that ablation of Lgr5+ cells caused synapse loss in inner HCs and loss of outer HCs, resulting in profound hearing loss.

11) Results middle of page 20: Please clarify how the expression pattern of GLAST-CreER differs from the Lgr5-DTR expression pattern so that it is clear what cells are fate-mapped vs. killed. Also please clarify what percentage of GER cells are targeted by GLAST-CreER with the P1 tamoxifen injection. Finally, Mellado Lagarde et al., 2014 and McGovern et al.., 2019 described the GLAST-CreER expression pattern with tamoxifen given at similar age, please add these citations.

12) Results middle of page 21 and Figure 6A-D: The Sparcl1 data seems out of place. Seems like it would fit better in Figure 1 or could be removed if you prefer. Just seems a weird fit for the translatomic figure. Also the GLAST-Cre expression pattern + Edu was just presented in Figure 5 - why is it shown again here?

13) Results page 22 & 23: What is nCounter measuring? Transcript level, protein levels, cells? And from what dataset - the RNAscope or something else? In Figure 6I-J and 7L, the nCounter data is presented as normalized counts. How/what is this normalized too? Details about nCounter and normalization need to be better explained and also added to the methods section.

14) Results top of page 24: cluster 2 genes are markers of HCs. This seems odd to me since the gene expression analysis was done using GLAST+ SCs and there is no evidence of HC regeneration after the Lgr5+ cells were ablated. Please add further explanation/speculation in the Discussion section for this.

15) There are several references to neonatal HC regeneration throughout the paper where Cox et al., 2014 is cited but another paper was published at the same time with similar findings (Bramhall et al., 2014) and should also be cited.

16) Discussion: Figure 4P-Q shows a loss of ribbon synapses at P21 in the Lgr5-DTR model and profound hearing loss by ABR and DPOAE in supplemental figures. However, Mellado Lagarde et al., 2014 saw no change in synapses and no hearing loss when inner phalangeal cells were killed using the PlpCre-DTA model. These differences should be addressed in the discussion.

17) Figure 2 legend: panels D' and E" are not orthogonal images just images of Edu and Ki67 without Sox2

18) Figure 3 legend: CD44 does not label Hensen cells. It labels Claudius cells and LER cells lateral to the Claudius cells (See Chrysostomou et al, 2020).

19) Figure 4 legend: For A, please add the age tamoxifen was injected as this Tomato pattern will change when TAM is injected at different ages.

Reviewer #3: The sensory epithelium of the mammalian cochlea contains sensory hair cells (HCs) and a variety of supporting cells (SCs). These cells exit the cell cycle during embryonic development and there is very limited capacity for cell regeneration after damage, except for a brief period after birth when both HCs and SCs can be regenerated. While most studies focus on HC regeneration, in this manuscript the authors focus on regeneration of SCs. The authors present a comprehensive study that provides new information on the regenerative capacity of inner ear cells; the findings are novel, advancing the field and the transcriptomic analysis forms the basis to explore the molecular mechanisms underlying this process.

The authors show that after genetic ablation of a subpopulation of SCs, cells in the greater epithelial ridge (GER) re-enter the cell cycle and replace medially located SCs (inner phalangeal cells: iPhCs), but not lateral SCs. They confirm the location and source of progenitors using lineage tracing together with cell ablation. They then characterise the regenerated iPhCs using molecular markers indicating that new cells are integrated in an interconnected network and assessing spontaneous calcium transients, as well as consequences on HC innervation. Overall, their results are consistent with regeneration of iPhCs from GER cells, although the regenerated cells do not fully resemble normal iPhCs. Finally, the authors use the RiboTag mouse to characterise the transcriptional profile of damaged and regenerated cells in comparison to controls and verify a number of differentially expressed and damage/regeneration response genes.

The authors' conclusions are generally well supported by the data presented including the supplementary information. However, the most results appear to be collected from the apex of the cochlea; only a few experiments evaluate mid- or basal turns. This is important since cells in the apex are the last to differentiate. The authors should show data from all three regions throughout all experiments, as customary in the field. If the regenerative capacity is indeed restricted to the apex, the findings cannot be generalised for the entire cochlea as suggested by the authors.

I suggest that the authors should reorganise the text and discuss their lineage tracing experiments before the characterisation of regenerated iPhCs. Page 5 of the main text, line 17 the authors conclude: "Collectively, these results indicate that … damage-activated proliferative cells migrate laterally to replace lost IPhCs." This conclusion is not justified by the data presented so far, but can be substantiated by the lineage tracing experiments.

---

## [Decision Letter · Decision Letter 2]

6 Oct 2021

Dear Dr Cheng,

Thank you for submitting your revised Research Article entitled "Translatomic analysis of mitotic cochlear progenitors after ablation of Lgr5+ cells" for publication in PLOS Biology. I have now obtained advice from the original reviewers and have discussed their comments with the Academic Editor. 

The reviews are appended below. As you will see, both Reviewers 2 and 3 are satisfied by the revision and note that the manuscript has been much improved. However, Reviewer 1 has a number of lingering concerns which will need to be addressed with further clarifications. S/he notes that the study does not adequately distinguish between damage or regeneration responses and that without testing candidate genes for regeneration the study lacks mechanistic insights. While we appreciate that the additional mechanistic studies suggested by Reviewer 1 would be interesting, after a careful discussion within the team and with the Academic Editor, we do not think that these analyses would be required for publication in PLOS Biology at this stage.

Therefore, we will probably accept this manuscript for publication, provided you satisfactorily address the remaining points raised by the Reviewer 1 with additional clarifications and a careful revision to ensure that the data is carefully interpreted and the conclusions are fully supported. **IMPORTANT: Please also make sure to address the following data and other policy-related requests.

1) DATA REQUEST: Please provide, as a supplementary file or deposition in a publicly available repository, the data underlying each figure in your manuscript. Note that we do not require all raw data. Rather, we ask that all individual quantitative observations that underlie the data summarized in the figures and results of your paper be made available. The numerical data provided should include all replicates AND the way in which the plotted mean and errors were derived (it should not present only the mean/average values).

Please ensure that you provide the individual numerical values that underlie the summary data displayed in the following figure panels as they are essential for readers to assess your analysis and to reproduce it:

Fig 1O-P; Fig 2H,O; Fig 3I,O; Fig 4F-H; Fig 5K-L,S-T; Fig 6G-H; Fig 7A-L; Fig S2M-N; Fig S3M-P; Fig 4H-I; Fig S5I-K; Fig S6I-J; Fig S7A; Fig S8A-D

More information on PLOS' Data Policy, which requires that all data be made available without restriction, can be found here: http://journals.plos.org/plosbiology/s/data-availability. For more information, please also see this editorial: http://dx.doi.org/10.1371/journal.pbio.1001797 

2) TITLE: After a bit of discussion within the team, we think that the title could be edited slightly to further highlight the advance of the study. If you agree, we would suggest that you change it to something like "Lineage-tracing and translatomic analysis of damage inducible mitotic cochlear progenitors identifies candidate genes regulating regeneration"

We expect to receive your revised manuscript within two weeks. 

*Published Peer Review History*

*Early Version*

Sincerely,

Lucas Smith, Ph.D.,

Associate Editor,

lsmith@plos.org,

PLOS Biology

Reviewer remarks:

Reviewer #1: This is a revision for the resubmission of a manuscript on the regenerative potential of an Lgr5-independent cell population in the GER. By disrupting the Lgr5 population, they show that the GER houses competent cells. 

From the perspective of regeneration, the findings in the paper are not new. The study includes translatome analyses, but the manuscript does little with this information to highlight any mechanistic understanding, which is what is lacking in the field. The study does not clearly distinguish between damage or regeneration responses. The way sections are written (abstract, introduction), the reader is misled to think that the differentially expressed genes are associated with regeneration until they get to the discussion, which is a little late. The authors will want to consider approaches to overexpress candidate genes for regeneration to support the regeneration pitch of this manuscript.

Despite CtBP2 synapses being drastically (quantitatively too) reduced, in the response letter, the authors claim that the IHC remain innervated? I'm not sure how that conclusion was reached. 

At this point, I cannot agree that the Ca transients are associated with regeneration (as suggested in the abstract). Thus far, most of the data suggest damage responses (despite several pieces of data have been removed), other than the one regeneration phenotype that has been described by several others. Other than ruling out a marker of interest (Lgr5), as mentioned before, it has been a long-known that the GER holds such a potential.

The authors need to carefully interpret or reevaluate their narration and conclusions, as it will cause a lot of confusion. 

Reviewer #2, Brandon C. Cox: Thank you for the thorough response to the reviewer comments. The manuscript is much improved and now ready for publication.

Reviewer #3: The authors have addressed previous comments and improved the manuscript.

---

## [Editor Report · Decision Letter 3]

18 Oct 2021

Dear Dr Cheng,

On behalf of my colleagues and the Academic Editor, Marianne Bronner, I am pleased to say that we can in principle offer to publish your Research Article "Lineage-tracing and translatomic analysis of damage inducible mitotic cochlear progenitors identifies candidate genes regulating regeneration" in PLOS Biology, provided you address any remaining formatting and reporting issues. These will be detailed in an email that will follow this letter and that you will usually receive within 2-3 business days, during which time no action is required from you. Please note that we will not be able to formally accept your manuscript and schedule it for publication until you have made the required changes.

PRESS

Sincerely, 

Lucas Smith, Ph.D. 

Senior Editor 

PLOS Biology

lsmith@plos.org